# Understanding MLP-Mixer as a Wide and Sparse MLP

## Abstract

Multi-layer perceptron (MLP) is a fundamental component of deep learning that has been extensively employed for various problems. However, recent empirical successes in MLP-based architectures, particularly the progress of the MLP-Mixer, suggest that our understanding of how MLPs achieve better performance remains limited and there is an underlying mechanism. In this research, we reveal that the MLP-Mixer effectively behaves as a wide MLP with sparse weights. Initially, we clarify that the mixing layer of the Mixer has an effective expression as a wider MLP whose weights are sparse and represented by the Kronecker product. It is also regarded as an approximation of Monarch matrices. Next, we confirmed similarities between the mixer and the unstructured sparse-weight MLP in hidden features and performance when adjusting sparsity and width. To verify the similarity in much wider cases, we introduced the RP-Mixer, a more memory-efficient alternative to the unstructured sparse-weight MLP. Then we verified similar tendencies between the MLP-Mixer and the RP-Mixer, confirming that the MLP-Mixer behaves as a sparse and wide MLP, and that its better performance is from its extreme wideness. Notably, when the number of connections is fixed and the width of hidden layers is increased, sparsity rises, leading to improved performance, consistent with the hypothesis by Golubeva, Neyshabur and Gur-Ari (2021). Particularly, maximizing the width enables us to quantitatively determine the optimal mixing layer's size.

## 1 Introduction

Multi-layer perceptron (MLP) and its variants are fundamental components of deep learning employed in various problems and for understanding the basic properties of neural networks. Despite their simplicity and long history (Schmidhuber, 2015), it has become apparent only recently that there is still significant room for improvement in the predictive performance of MLP-based architectures. Some studies have developed learning or pruning algorithms to obtain better weights (Neyshabur, 2020; Pellegrini & Biroli, 2022; d'Ascoli et al., 2019) while others have proposed some modifications to the network architectures (Golubeva et al., 2021; Tolstikhin et al., 2021; Touvron et al., 2022). The latter line of study, which the current study follows, includes thought-provoking empirical facts.

Golubeva et al. (2021) reported that the prediction performance improves by increasing the width and the sparsity of connectivity when the number of trainable parameters is fixed. In other words, under appropriate control of the parameter size, a wide sparse MLP works better than its dense counterpart. The MLP-Mixer is another noteworthy example of recent developments in MLP-based architectures (Tolstikhin et al., 2021). It does not rely on convolutions or self-attention and is entirely composed of MLP layers; instead, it utilizes MLPs applied across spatial locations or feature channels. This can be regarded as an MLP that applies fully connected (FC) layers on both sides of the feature matrix. Despite its simplicity, the MLP-Mixer achieved a performance on image classification benchmarks comparable to that of more structured deep neural networks.

While the performance of MLP-based architectures has improved, our understanding of which factors help this performance improvement remains limited. Currently, the MLP-Mixer is expected to achieve the highest accuracy among MLP-based architectures. However, there are few studies on experimental and theoretical attempts to understand its internal mechanisms (Yu et al., 2022; Sahiner et al., 2022). To further advance the MLP-based architecture, it is crucial to address questions such as at what point the mixer differs from a conventional MLP and which settings contribute to its superior performance.

In this study, we reveal some underlying mechanics of the MLP-Mixer and contribute to enriching our understanding of its appropriate design. We find that a wide and sparse MLP (Golubeva et al., 2021) and the MLP-Mixer, seemingly unrelated MLP variants, are essentially related, and this relationship plays a fundamental role in understanding the inner workings of the MLP-Mixer. The contributions of this study are summarized as follows:

- We show *an effective expression of MLP-Mixer as an MLP* by vectorizing the mixing layers (in Section 3). It is composed of the permutation matrix and the Kronecker product and provides an interpretation of mixing layers as an extremely wide MLP with sparse (structured) weights. Furthermore, a certain MLP-Mixer can be regarded as an approximation of MLP with the Monarch matrix (in Section 3.3).

- We found similar tendencies between the mixer and unstructured sparse-weight MLP in hidden features and the performance under widening with a fixed number of connections. For wider sparse-weight MLPs, the unstructured approach becomes impractical due to its extensive memory and computational demands. As a solution, we introduce the *RP-Mixer*, a memory-efficient, computationally undemanding, and lightly structured alternative (in Section 4.2). Then, we confirmed the MLP-Mixer and the alternative have similar tendencies in wider cases (in Section 5).

- Before introducing the RP-Mixer, we characterize the Mixers as a special example of a general class: *Permuted-Kronecker (PK) family* (in Section 4.1). Based on the hypothesis by Golubeva et al. (2021), with maximizing the widths in the PK family, we find the appropriate sizes of channel and token mixing layers (in Section 5.1 & 5.2). Therefore, this work provides not only a qualitative understanding but also quantitative implications that can help architecture designs.

## 2 PRELIMINARIES

### 2.1 RELATED WORK

**MLP-based architectures.** The salient property of an MLP-Mixer is that it is composed entirely of FC layers. This property is unique to the MLP-Mixer (and its concurrent work ResMLP (Touvron et al., 2022)) and different from attention-based architectures (Dosovitskiy et al., 2021). While some previous work focused on providing a relative evaluation of performance compared with the attention module (Yu et al., 2022; Sahiner et al., 2022), our purpose is to understand the essential function of the MLP-Mixer and to provide a novel characterization as a wide and sparse MLP. Golubeva et al. (2021) investigated that the generalization performance can be improved by increasing the width in FC layers. Because they fixed the number of weights, an increase in the width caused a higher sparsity. They revealed that even for fixed sparse connectivity throughout training, a large width can improve the performance better than the dense layer. As the MLP-Mixer is expressed by an extremely wide MLP with certain sparse weights, we can apply their argument to the mixing layers and determine the appropriate layer size.

**Structured weight matrices: (i) Sparse matrix.** Parameter sparsity is widely used to improve the performance and efficiency of neural networks. One approach is to determine nonzero weights dynamically, such as dense-to-sparse training (Neyshabur, 2020), pruning (Frankle & Carbin, 2019), and sparse-to-sparse training (Dettmers & Zettlemoyer, 2019; Evci et al., 2020; Liu et al., 2022). The other is to constrain the trainable weights from the beginning of training. statically (Dao et al., 2022; Golubeva et al., 2021). The current study follows the latter approach; specifically, we reveal that the mixing layers of the MLP-Mixer are implicitly related to such fixed sparse connectivity. **(ii) Kronecker product.** Constraining weight matrices to the Kronecker product and its summation has been investigated in the model-compression literature. Some works succeeded in reducing the number of learnable parameters without deteriorating the prediction performance (Zhou et al., 2015; Zhang et al., 2021) while others applied them for the compression of trained parameters (Hameed et al., 2022). In contrast, we find a Kronecker product expression hidden in the MLP-Mixer, which can be regarded as an approximation of the Monarch matrix proposed in Dao et al. (2022).

Notably, our study is completely different from simply applying the sparsity of Golubeva et al. (2021), Kronecker-product weights, or Monarch matrices to the dense weight matrices of the mixing layers. Our finding is that effective MLP expression and its generalization (i.e., the PK family) inherently possess these properties.

## 2.2 NOTATIONS

**MLP-Mixer**   An MLP-Mixer is defined as follows (Tolstikhin et al., 2021). Initially, it divides an input image to patches. Next, a per-patch FC layer is performed. After that, the blocks described as follows are repeatedly applied to them: for the feature matrix from the previous hidden layer $X \in \mathbb{R}^{S \times C}$,

$$\text{Token-MLP}(X) = W_2\phi(W_1 X), \quad \text{Channel-MLP}(X) = \phi(X W_3) W_4, \tag{1}$$

where $W_1 \in \mathbb{R}^{\gamma S \times S}$, $W_2 \in \mathbb{R}^{S \times \gamma S}$, $W_3 \in \mathbb{R}^{C \times \gamma C}$ and $W_4 \in \mathbb{R}^{\gamma C \times C}$. In this paper, we set the expansion factor of the hidden layers of token and channel-mixing MLPs to the same value $\gamma$ for simplicity. The block of the MLP-Mixer is given by the map $X \mapsto Y$, where

$$U = X + \text{Token-MLP}(\text{LN}(X)), \;\; Y = U + \text{Channel-MLP}(\text{LN}(U)). \tag{2}$$

In the end, the global average pooling and the linear classifier are applied to the last hidden layer.

**Remark on per-patch FC layer:**   The first layer of the MLP-Mixer is given by the so-called per-patch FC layer, which is a single-layer channel mixing. The input image is divided into $S_0$ patches of size $C_0$, and the original study set the size of the mixing layers to $(S, C) = (S_0, C_0)$. In contrast, to investigate the contribution of each input image size and hidden layer size independently, it is rational to change $(S, C)$ independent of $(S_0, C_0)$. Therefore, we make the per-patch FC transform the input size $C_0$ to the output size $C$ and the first token mixing layer transform $S_0$ to $S$. A more detailed formulation of the architectures is presented in Section A of the Supplementary Material. Our results hold, even for $(S, C) = (S_0, C_0)$ as discussed later.

## 3   SIMILARITY BETWEEN MLP-MIXER AND MLP VIA VECTORIZATION

To address the similarity between MLP-Mixer and MLP, we consider vectorization of feature tensors and effective width.

**Vectorization and effective width:**   We represent the vectorization operation of the matrix $X \in \mathbb{R}^{S \times C}$ by $\text{vec}(X)$; more precisely, $(\text{vec}(X))_{(j-1)d+i} = X_{ij}, (i = 1, \ldots, S, j = 1, \ldots, C)$. We also define an inverse operation $\text{mat}(\cdot)$ to recover the matrix representation by $\text{mat}(\text{vec}(X)) = X$. There exists a well-known equation for the vectorization operation and the Kronecker product denoted by $\otimes$;

$$\text{vec}(WXV) = (V^\top \otimes W)\text{vec}(X), \tag{3}$$

for $W \in \mathbb{R}^{S \times S}$ and $V \in \mathbb{R}^{C \times C}$. The vectorization of the feature matrix $WXV$ is equivalent to a fully connected layer of width $m := SC$ with a weight matrix $V^\top \otimes W$. We refer to this $m$ as the *effective width* of mixing layers.

In MLP-Mixer, when we treat each $S \times C$ feature matrix $X$ as an $SC$-dimensional vector $\text{vec}(X)$, the right multiplication by an $C \times C$ weight $V$ and the left weight multiplication by a $S \times S$ weight $W$ are represented as follows by (3):

$$\text{vec}(XV) = (V^\top \otimes I_S)\text{vec}(X), \;\; \text{vec}(WX) = (I_C \otimes W)\text{vec}(X). \tag{4}$$

This expression clarifies that the mixing layers work as an MLP with special weight matrices with the Kronecker product. As usual, the size of $S$ and $C$ is approximately $10^2 \sim 10^3$, and this implies that the Mixer is equivalent to an extremely wide MLP with $m = 10^4 \sim 10^6$. Moreover, the ratio of non-zero entries in the weight matrix $I_C \otimes W$ is $1/C$ and that of $V^\top \otimes I_S$ is $1/S$. Therefore, the weight of the effective MLP is highly sparse.

Furthermore, to consider only the left-multiplication of weights, we introduce commutation matrices.

**Commutation Matrix:**   A *commutation matrix* $J_c$ is defined as

$$J_c\text{vec}(X) = \text{vec}(X^\top), \tag{5}$$

where $X$ is an $S \times C$ matrix. Note that for any $x \in \mathbb{R}^m$ and entry-wise function $\phi$, $J_c\phi(x) = \phi(J_c x)$. In addition, we have $J_c^\top(I_S \otimes V^\top)J_c = V^\top \otimes I_S$ for any $C \times C$ matrix $V$.

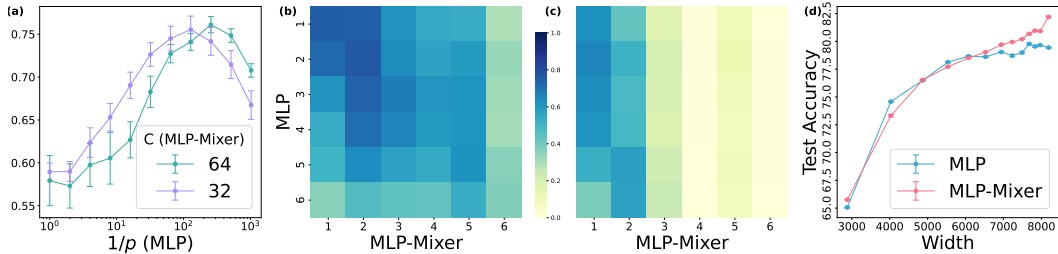

Figure 1: (a) Average of diagonal entries of CKA between trained MLP-Mixer ($S = C = 64, 32$) and MLP with different sparsity, where $p$ is the ratio of non-zero entries in $M$. (b) CKA between MLP-Mixer ($S = C = 64$) and MLP with the corresponding $p = 1/64$, and (c) CKA between the Mixer and a dense MLP. (d) Test accuracy of MLPs with sparse weights and MLP-Mixers with different widths $\gamma m$ under the fixed number of connections per layer $\Omega = 2^{19}$. Each experiment is done on CIFAR10 with four random seeds.

**Case of S-Mixer:** As a starting point, let us introduce an *S-Mixer*, which is a simple version of the MLP-Mixer, by removing hidden layers of mixing MLPs:

$$\text{Token-Layer}(X) = \phi(WX), \quad \text{Channel-Layer}(X) = \phi(XV). \tag{6}$$

We find that the following expression holds for the mixing layers:

**Proposition 3.1** (Effective expression of MLP-Mixer as MLP). *Suppose an S-Mixer for simplicity. A feature matrix $H = \phi(\phi(WX)V) \in \mathbb{R}^{S \times C}$ is expressed as a shallow MLP with width $m = SC$ using the Kronecker products:*

$$vec(H) = \phi \left( J_c^\top \left( I_S \otimes V^\top \right) \phi \left( J_c \left( I_C \otimes W \right) vec \left( X \right) \right) \right). \tag{7}$$

The derivation is straightforward, as described in Section B.1. This expression clarifies that the mixing layers work as an MLP with special weight matrices with the commutation matrix and Kronecker product.

**Case of MLP-Mixer:** It is easy to generalize the above expression for the S-Mixer to a standard MLP-Mixer, where each mixing operation is composed of shallow neural networks (1). The MLP-Mixer (2) is expressed as follows: $Y = \text{Channel-MLP}(\text{Token-MLP}(X)))$

$$u = \phi(J_c(I_C \otimes W_2)\phi((I_C \otimes W_1)x)), \quad y = \phi(J_c^\top(I_S \otimes W_4^\top)\phi((I_S \otimes W_3^\top)u)) \tag{8}$$

where $u = \text{vec}(U)$ and $y = \text{vec}(Y)$. This equivalence with a wide MLP with sparse weights is simple and easy to follow but has been missing in the literature. One can regard this as an implicit bias behind Mixers.

### 3.1 COMPARING HIDDEN FEATURES

To validate the similarity of networks in a robust and scalable way, we look at the similarity of hidden features of MLPs with sparse weights and MLP-Mixers based on the centered kernel alignment (CKA) (Nguyen et al., 2021). In practice, we computed the mini-batch CKA (Nguyen et al., 2021, Section 3.1(2)) among features of trained networks. Detailed settings of all experiments are summarized in Section. C.

In the literature on sparse weights, a naive implementation is to set a random sparse mask (Golubeva et al., 2021). We denote by *unstructured sparse-weight MLP* (SW-MLP in short) an MLP whose weights are replaced by $M \odot A$ where $A$ is a parameter matrix and $M$ is a static mask matrix whose entries are drawn from the Bernoulli distribution with a probability $p > 0$ of being one at the initialization phase. In this section, we consider a SW-MLP constructed by replacing $I_C \otimes W$ (resp. $V^\top \otimes I_S$) of an MLP-Mixer by $M \odot A$ where $A$ is a full parameter matrix and $M$ is the static mask with $p = 1/C$(resp. $1/S$). In Fig. 1(a), we observed the averaged CKA achieved the maximum as an appropriate sparsity of the MLPs. By comparing Fig. 1(b) and (c), we found that CKA matrix with sparse MLP was clearly higher than dense MLP. In particular, the sparse Mixer was similar to sparser MLP in hidden features.

### 3.2 COMPARING ACCURACY WITH INCREASING EFFECTIVE WIDTH

To validate the similarity, we compare the test accuracy of both networks with different sparsity. Under the fixed number of connectivity per layer, denoted by $\Omega$, the sparsity is equivalent to the wideness. Fig.1(d) shows the test accuracy of MLP-Mixers and corresponding sparse weight MLPs under fixed $\Omega = 2^{19}$ and $\gamma = 2$, for several widths $\gamma m$. We observed both networks' test accuracy increased as width increased. Note that we observed the test accuracy of both networks decreased as width (sparsity) increased except for too-wide cases around $\gamma m = 8000$ in Fig.1(d). Golubeva et al. (2021) reported that if the sparsity became too high, the generalization performance of SW-MLP slightly decreased. They discussed that this decrease was caused by the deterioration of trainability, that is, it became difficult for the gradient descent to decrease the loss function. This seems rational because of the spectrum of the weight $M \odot A$ and $I_C \otimes W$ (or $V^\top \otimes I_S$) are different. We discuss on the spectrum in Section D.1. In conclusion, MLP and MLP-Mixer have a similar tendency for the performance with increasing width, except in extreme cases.

### 3.3 MONARCH MATRIX HIDDEN BEHIND MIXERS

Dao et al. (2022) proposed a *Monarch matrix* $M \in \mathbb{R}^{n \times n}$ defined by

$$M = J_c^\top L J_c R, \tag{9}$$

where $L$ and $R$ are the trainable block diagonal matrices, each with $\sqrt{n}$ blocks of size $\sqrt{n} \times \sqrt{n}$. The previous work claimed that the Monarch matrix is sparse in that the number of trainable parameters is much smaller than in a dense $n \times n$ matrix. Despite this sparsity, by replacing the dense matrix with a Monarch matrix, it was found that various architectures can achieve almost comparable performance while succeeding in shortening the training time. Furthermore, the product of a few Monarch matrices can represent many commonly used structured matrices such as convolutions and Fourier transformations.

Surprisingly, the MLP-Mixer and Monarch matrix, two completely different concepts developed in the literature, have hidden connections. By comparing (7) and (9), we find that

**Corollary 3.2.** *Suppose a block of S-Mixer $H$ with $C = S$ whose activation function in the intermediate layer is linear, that is, $H = \phi(WXV)$. Then, $vec(H)$ is equivalent to an MLP whose weight matrix is given by a Monarch matrix with weight-sharing diagonal matrices, that is, $\phi(Mx)$ with $n = SC$, $L = I_S \otimes V^\top$ and $R = I_C \otimes W$.*

In Section B.4, we additionally discuss the Monarch matrix.

## 4 ALTERNATIVE TO SPARSE WEIGHT MLP

As seen in Section 3, MLP-Mixers have similar tendencies to wide and sparse MLPs. In much wider models, to continue comparing MLP-Mixer and unstructured sparse-weight MLP, we need an alternative to static-masked MLP because of its huge computational costs and memory requirements. Thus we further discuss on structure of MLP-Mixer with its effective expression, and then, by destroying a part of it, we introduce an alternative model of static-masked MLP.

### 4.1 PK FAMILY

To introduce an alternative to sparse-weight MLPs, we propose a permuted Kronecker (PK) family as a generalization of the MLP-Mixer.

**Permutation matrix:** An $m \times m$ permutation matrix $J$ is a matrix given by $(Jx)_i = x_{\sigma(i)}, (i = 1, 2, \ldots, m)$ for an index permutation $\sigma$. In particular, the commutation matrix $J_c$ is a permutation matrix (Magnus & Neudecker, 2019). Note that for any permutation matrix $J$, $x \in \mathbb{R}^m$ and entry-wise function $\phi$, $J\phi(x) = \phi(Jx)$.

**Definition 4.1** (PK layer and PK family). *Let $J_1, J_2$ be $m \times m$ permutation matrices. For $X \in \mathbb{R}^{n_1 \times n_2}$, we define the PK layer as follows:*

$$PK\text{-}Layer_W(X; J_1, J_2) := \phi[J_2(I_{n_1} \otimes W)J_1 vec(X)], \tag{10}$$

| Model | Memory(CIFAR/ImageNet) | FLOPs(CIFAR/ImageNet) | Runtime(CIFAR) |
|---|---|---|---|
| SW-MLP | 3.22GB / 23.2TB | 805M / 5.80T | $28.2(\pm0.00)$ s/epoch |
| RP-Mixer | 3.94MB / 26.3MB | 12.5M / 4.06G | $6.41(\pm0.01)$ s/epoch |
| MLP-Mixer | 3.93MB / 26.3MB | 12.5M / 4.06G | $6.08(\pm0.26)$ s/epoch |

Table 1: Comparison on memory requirements, FLOPs(floating point operations), and averaged runtime. For the three models, we set $S = 256, C = 588, L = 8, \gamma = 4$ for ImageNet and $S = 64, C = 48, L = 3, \gamma = 2$ for CIFAR.

*where we set $m = n_1 n_2$, $W \in \mathbb{R}^{n_2 \times n_2}$ and $I_n$ denotes an $n \times n$ identity matrix. We refer to the set of architectures whose hidden layers are composed of PK layers as the PK family.*

Since $J_c$ is a permutation matrix, the normal S-Mixer and MLP-Mixer belong to the PK family. (See Section B.3 for the details.) The important point of the PK-Layer is that its width $m$ is possibly large, but there is *no need to explicitly preserve the $m \times m$ weight matrix* in memory. We can compute the forward signal propagation by a relatively small matrix multiplication in the same manner as the MLP-Mixer: First, $J_1 \text{vec}(X) =: y$ is a rearrangement of $X$ entries. Next, we compute pre-activation by using the matrix product $(I_{n_1} \otimes W)y = W\text{mat}(y)$. Finally, we apply entry-wise activation and rearrangement by $J_2$. Thus, the PK layer is memory-friendly, whereas the naive dense MLP requires preserving an $m \times m$ weight matrix and is computationally demanding.

## 4.2 RANDOM PERMUTED MIXERS

In normal Mixers, $J_1$ and $J_2$ are restricted to the identity or commutation. This means that the sparse weight matrices of the effective MLP are highly structured because their block matrices are diagonal. To destroy the structure, we introduce RP-Mixers. A *RP S-Mixer* has $(J_1, J_2)$ in each PK layer, which is given by random permutation matrices as $U = \text{PK-Layer}_W(X; J_1, J_2)$ and $\text{PK-Layer}_{V^\top}(U; J_1', J_2')$. Similarly, for a *RP MLP-Mixer*, we set the PK layer corresponding to token mixing and channel mixing to the random permutation matrices. From (10), this is equivalent to the effective MLP with width $SC$ (and $\gamma SC$ for the MLP-Mixer) and sparse weight

$$W_{eff} = J_2(I_{n_1} \otimes W)J_1. \tag{11}$$

Because $(J_1, J_2)$ are random permutations, the non-zero entries of $W_{eff}$ are scattered throughout the matrix. In this sense, RP Mixers seemingly become much closer to random sparse weights than the normal Mixers. As a technical remark, we can implement RP Mixers without skip connections more simply by using a random $J_2$ and setting $J_1 = I$. This is because the product of $J_2$ in the current layer and $J_1$ in the next layer is also an instance of a random permutation (See Section B.2).

Table 1 shows the computational resource of SW-MLPs and RP-Mixers. In the setting for ImageNet, SW-MLP requires huge memory and runtime, whereas RP-Mixer needs $10^3$ to $10^6$ times less. Note that the spacial complexity of the SW-MLP (resp. the RP-Mixer) is $O(m^2) = O(S^2 C^2)$ (resp. $O(C^2 + S^2)$) as $S, C \to \infty$. Therefore, the RP-Mixer is more memory-efficient and computationally undemanding than the SW-MLP.

## 5 REVISIT THE SIMILARITY IN WIDER CASES

In this section we continue the discussion on whether the MLP-Mixer (or S-Mixer) exhibits tendencies similar to those of unstructured (or lightly-structured) sparse-weight MLPs in wider cases than Section 3. The following hypothesis has a fundamental role:

**Hypothesis 5.1** (Golubeva et al. (2021)). *An increase in the width while maintaining a fixed number of weight parameters leads to an improvement in test accuracy.*

Intuitively, Golubeva et al. (2021) challenged the question of whether the performance improvement of large-scale deep neural networks was due to an increase in the number of parameters or an increase in width. They empirically succeeded in verifying Hypothesis 5.1; that is, the improvement was due to the increase in width in normal MLPs and ResNets (note that the width of ResNet indicates the channel size).

| MLP | CIFAR-10 | CIFAR-100 | max. width | #connections |
|---|---|---|---|---|
| $\beta$-LASSO (Neyshabur, 2020) | 85.19 | 59.56 | - | 256M |
| Mixer-SS/8 | 84.09 ($\pm$1.55) | 55.76($\pm$1.83) | $6.3 \times 10^4$ | 256M |
| Ours | **87.93** ($\pm$0.47) | **61.87**($\pm$1.36) | $\mathbf{1.2 \times 10^5}$ | 255M |

Table 2: Test accuracy on CIFAR-10/100 from scratch. By setting $S$ and $C$ closer **under the same number of total connections** throughout layers, the maximal width of layers became larger in ours. Its test accuracy eventually improved than $\beta$-LASSO. Each experiment is done with three random seeds.

| MLP | ImageNet-1k | max. width | $\Omega$ | S | C |
|---|---|---|---|---|---|
| Mixer-B/16Tolstikhin et al. (2021) | 76.44 | $6.0 \times 10^5$ | $2.6 \times 10^8$ | 196 | 786 |
| Ours | **76.74** ($\pm$ 0.19 ) | $\mathbf{6.2 \times 10^5}$ | $2.6 \times 10^8$ | 256 | 588 |

Table 3: Test accuracy on ImageNet-1k from scratch. Experiments of our model are done with three random seeds.

Here, for the MLP-Mixer, the average number of non-zero entries per layer (denoted by $\Omega$) is given by

$$\Omega = \gamma(CS^2 + C^2S)/2. \tag{12}$$

From (11), the number of non-zero entries in the PK layer is $n_1 n_2^2$. This means that for a block of the S-Mixer (7), the average number of non-zero entries per layer (denoted by $\Omega$) is given by $\Omega = (CS^2 + C^2S)/2$. The average number $\Omega$ of S-Mixer is reduced to $\gamma = 1$ in (12), which maintains the readability of the equations.

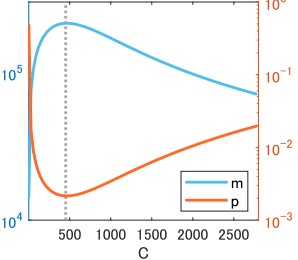

Figure 2: Theoretical line of $m$ and $p$ ($\Omega = 10^8, \gamma = 1$)

By (12), we have $S = (\sqrt{C^2 + 8\Omega/(\gamma C)} - C)/2$. For a fixed $\Omega$ and $\gamma$, the effective width is controlled by $m = SC$. Fig. 2 shows $m$ as a function of $C$. The width $m$ has a single-peak form and is maximized at $(C^*, S^*)$ as follows:

$$C^* = S^* = (\Omega/\gamma)^{1/3}, \quad \max_{S,C} m = (\Omega/\gamma)^{2/3}. \tag{13}$$

The ratio of non-zero entries $p := \Omega/m^2$ is minimized at this point, that is, the sparsity is maximized.

## 5.1 INCREASING WIDTH

Fig. 3 shows that the test accuracy improves as the effective width of the Mixers increases. We trained the normal and RP S-Mixers for various values of $S$ and $C$ with fixed $\Omega$. The normal and RP S-Mixers show similar tendencies of increasing test accuracy with respect to the effective width $m$. The normal and RP MLP-Mixers also show similar tendencies as is shown in Section D.1. Because the static-mask SW-MLP requires an $m \times m$ weight matrix, an SW-MLP with a large $\Omega$ cannot be shown in the figure. In contrast, we can use a large effective width for the Mixers, and the accuracy continues to increase as the width increases. This can be interpreted as the Mixers realizing a width setting where naive MLP cannot reach sufficiently. Eventually, the figure suggests that such an extremely large width is one of the factors contributing to the success of the Mixer.

Table 2 shows a comparison of MLP-Mixer with a dynamic sparsity $\beta$-LASSO (Neyshabur, 2020). We found a wider Mixer (Ours) has better performance than $\beta$-LASSO. Table 3 shows a comparison of Mixer-B/16(Tolstikhin et al., 2021) and a wider Mixer (Ours). In both results, the wideness improved the performance even if $\Omega$ is fixed. We observed in Table 3, by setting $S$ and $C$ closer under fixed $\Omega$, the maximal width of layers became larger in ours. Its test accuracy eventually improved more than the original MLP-Mixer B/16.

## 5.2 PERFORMANCE AT THE HIGHEST WIDTH AND SPARSITY

Fig. 4 confirms that the maximum width (13) derived from Hypothesis 5.1 adequately explains the empirical performance of the Mixers. Models were trained using supervised classifications for each

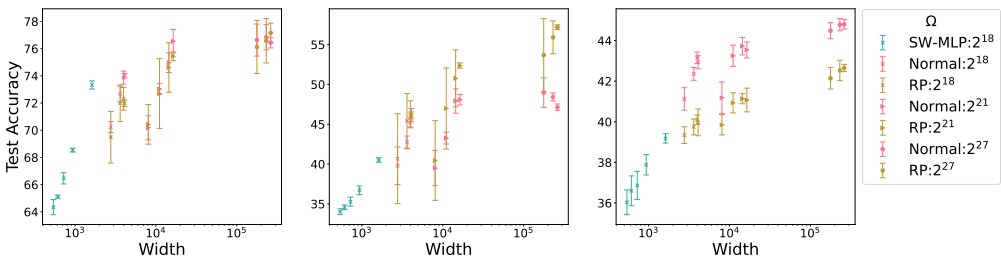

Figure 3: Test accuracy improves as the effective width increases. S-Mixer, RP S-Mixer and SW-MLP on (Left) CIFAR-10, (Center) CIFAR-100, (Right) STL-10.

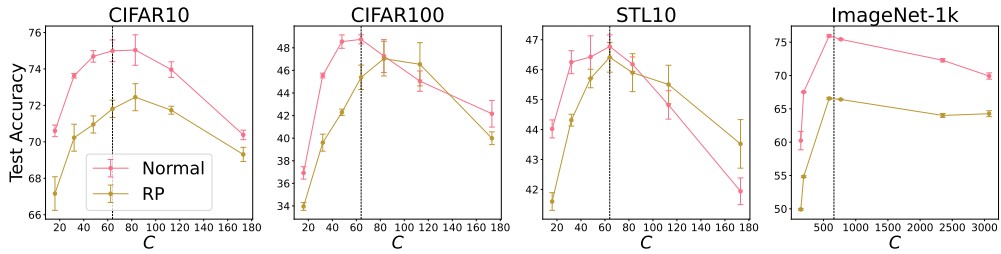

Figure 4: PK family achieves the highest test accuracy around $C = S$. S-Mixers on (a) CIFAR-10, (b) CIFAR-100, (c) STL-10. MLP-Mixers on (d) ImageNet-1k. Red line: Normal Mixers, yellow line: RP Mixers, dashed lines: $C = S$.

dataset. For CIFAR-10, CIFAR-100 and STL-10, we trained normal and RP S-Mixers. We fixed the dimension of the per-patch FC layer and changed $S$ and $C$ while maintaining a fixed number of $\Omega$. It can be observed that the test accuracy is maximized around $C = S$, as expected from Hypothesis 5.1 and (13). This tendency was common to the normal and RP Mixers. For ImageNet-1k, we trained normal and RP MLP-Mixers. Similarly, its performance is maximized around $C = S$.

## 5.3 INCREASING EXPANDING FACTOR

The effective MLP expression of the MLP-Mixer (8) has two widths: $m = SC$ and $\gamma SC$. As both are proportional to $SC$, we gave focused on changing $SC$ and fixed $\gamma$ so far. Here, we consider a complementary setting, that is, changing $\gamma$ with fixed $m = SC$. By substituting $S = m/C$ into (12), we obtain $\gamma = 2\Omega/(m(C + m/C))$. Similar to $m$ of $C$ shown in Fig. 2, this $\gamma$ is a single-peak function of $C$ and takes its maximum as

$$C^* = S^* = \sqrt{m}, \quad \max_{S,C} \gamma = \Omega/(m\sqrt{m}). \tag{14}$$

Fig. 5(left) confirms that increasing the width ($\gamma$) leads to performance improvement as is expected from Hypothesis 5.1. We trained normal and RP MLP-Mixers with various $\gamma$ in a realistic range. We plotted some cases of fixed $\Omega$ under the same $m$. Fig. 5(right) shows the test accuracy maximized around $C = S$ as is expected from (14).

## 5.4 DEPENDENCE ON DEPTH

As shown in Figures 3-5, both the normal and RP Mixers exhibited similar tendencies for a fixed depth. Fig. 6 confirms that by increasing the depth, i.e., the number of blocks, RP S-Mixers can even become comparable to the normal ones or better than them in some cases. First, we observed that, when the depth was limited, the RP Mixers were inferior to the normal Mixers in most cases. As we increased the depth, we observed that in some cases, overfitting occurred for the normal Mixer, but not for the RP one (see also the training and test losses shown in Section D.6). In such cases, the results of the RP Mixers were comparable (in Figs. 6(left, right)) or better (in Fig. 6(center)). Although RP Mixers are not necessarily better than normal ones, it is intriguing that even RP Mixers defined by a random structure can compete with normal Mixers.

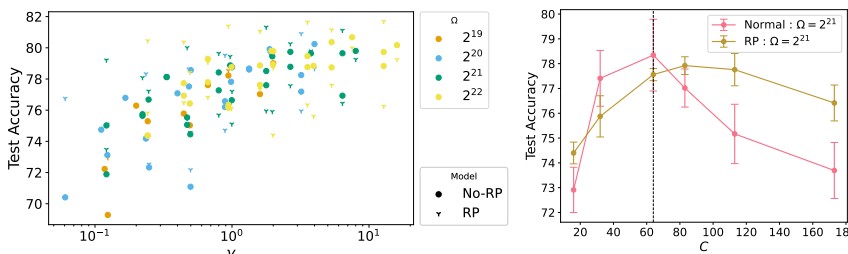

Figure 5: (Left) Increasing expansion factor $\gamma$ improves the test accuracy in normal and RP MLP-Mixers. (Right) The highest accuracy is achieved around $C = S$ with fixed $m = 4096$.

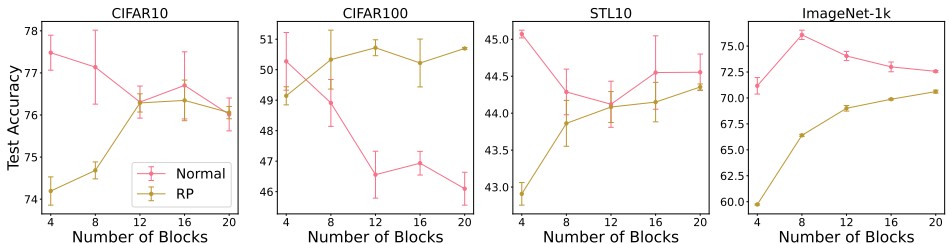

Figure 6: RP Mixers can become comparable to or even beat normal ones if the depth increases. We set $C = S = 128$.

## 6 CONCLUSION AND FUTURE WORK

This work provides novel insight that the MLP-Mixer effectively behaves as a wide MLP with sparse weights. The effective expression of the Mixer as the MLP elucidates the PK family as a more general class, RP-Mixers with more unstructured sparse weights, and hidden connections to the MLP with Monarch matrices. We empirically verified the similarities of MLP-Mixers with unstructured sparse and wide MLPs, or with RP-Mixers as an alternative in wider situations. Moreover, we empirically demonstrated that Mixers achieved higher prediction accuracy when the mixing layer sizes were set to increase the effective width of the Mixers. These results imply that one reason for the high performance of the Mixer is that it achieves an extremely large effective width through sparse connectivity. Thus, our work provides some quantitative suggestions on the appropriate design of mixing layers and a foundation for exploring further sophisticated designs of MLP-based architectures and the efficient implementation of neural networks.

In future work, it will be interesting to explore the possibility of other architectures or structured weights that can achieve a very large effective width, such as in the Mixers, while keeping the number of weights fixed. The MLP-Mixer and PK families allow matrix-form forward signal propagation, which is memory-friendly in that there is no need for naive preservation of a huge weight matrix. It would be interesting to clarify whether other potential candidates for memory-friendly architectures can achieve large effective widths. It would also be interesting to analyze the expressivity of MLPs with structured weights theoretically. In particular, because the effective weights of certain Mixers can be regarded as Monarch matrices approximated by weight sharing, evaluating the validity of such an approximation and the effects of Monarch products seems to be an interesting theme from the perspective of structural matrices.

**Reproducibility Statement:** We have detailed all experimental settings in Section C in the Appendix as Supplementary Materials. We also documented the number of random seeds and the GPUs used for the experiments. The claims made in the propositions within the main text are written in Section B. Additionally, the codes used for the experiments have been submitted as supplementary material. The experiments on ImageNet-1k were conducted using standard settings based on the publicly available library, pytorch-image-models (Wightman, 2019).

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

# Supplementary Materials

## A  DETAILS OF ARCHITECTURES

Here, we overview more technical details of all models: MLP-Mixer, Simple Mixer (S-Mixer), and MLP with sparse weights (SW-MLP). In Section A.1, we introduce the transformation from the input image to the first hidden layer. In Section A.2, we overview some detailed formulation of the models including skip connection and layer normalization.

### A.1  PER-PATCH FC LAYER

In all experiments, for a patch size $P$, the input image is decomposed into $HW/P^2$ non-overlapping image patches with size $P \times P$; we rearrange the $H \times W$ input images with 3 channels into a matrix whose size is given by $(HW/P^2) \times 3P^2 = S_0 \times C_0$. For the rearranged image $X \in \mathbb{R}^{S_0 \times C_0}$, the per-patch fully connected (FC) layer is given by

$$Y = XW^\top, \tag{S.1}$$

where $W$ is a $C \times C_0$ weight matrix. We use the per-patch FC layer not only for Mixers but also for SW-MLP.

### A.2  MLP-MIXER AND S-MIXER

Let us denote a block of the MLP-Mixer by

$$f_{W_1, W_2}(X) = \phi(XW_1^\top)W_2^\top, \tag{S.2}$$

and that of the S-Mixer by

$$f_{W_1}(X) = \phi(XW_1^\top). \tag{S.3}$$

We set $\phi = \text{GELU}$.

#### A.2.1  MLP-MIXER

We set the layer normalization (LN) by

$$\text{LN}(X) = \frac{X - m(X)}{\sqrt{v(X) + \epsilon}} \odot \gamma + \beta, \quad X \in \mathbb{R}^{S \times C}, \tag{S.4}$$

where $\odot$ denotes the Hadamard product , $m(X)$ (resp. $v(X)$) is the empirical mean (resp. the empirical variance) of $X$ with respect to the channel axis, and $\gamma, \beta$ are trainable parameters. We set $\epsilon = 10^{-5}$ in all experiments.

In the implementation of fully connected layers, we use only the right matrix multiplications in the same way as the original MLP-Mixer Tolstikhin et al. (2021). A token-mixing block $X \mapsto U$ of MLP-Mixer is given by

$$U = X + f_{W_1, W_2}(\text{LN}(X)^\top)^\top, \tag{S.5}$$

where $W_1$ is $S \times \gamma S$ and $W_2$ is $\gamma S \times S$. Similarly, we set a channel-mixing block $U \mapsto Y$ as

$$Y = U + f_{W_3, W_4}(\text{LN}(U)), \tag{S.6}$$

where $W_3, W_4$ are weight matrices.

We refer to the composed function $X \mapsto Y$ of the token-mixing block and the channel-mixing one as *a base block* of the MLP-Mixer. The MLP-Mixer with $L$-blocks is composed in the order of the per-patch FC layer, the $L$ base blocks, and the global average pooling with the layer normalization, and the last fully connected classification layer.

### A.2.2  S-MIXER

The S-Mixer without random permutations is implemented by replacing the MLP-block $f_{W_1,W_2}$ and $f_{W_3,W_4}$ in the MLP-Mixer with FC blocks. That is, token-mixing and channel-mixing blocks are given by

$$U = X + f_W(\text{LN}(X)^\top)^\top, \tag{S.7}$$
$$Y = U + f_V(\text{LN}(U)), \tag{S.8}$$

where $W$ and $V$ are weight matrices. The transpose of the input matrix in the token-mixing block is implemented by rearrangement of entries. We decided to apply both skip-connection and layer normalization even in the S-Mixer. This is a rather technical requirement for ensuring the decrease of training loss in deep architectures.

### A.2.3  MIXERS WITH PERMUTED KRONECKER LAYERS

Here we implement generalized MLP-Mixer and S-Mixer with permutation matrices and PK-layers. Recall that for any matrix $X$,

$$X^\top = \text{Mat}(J_c\text{vec}(X)), \tag{S.9}$$

where $J_c$ is the $m \times m$ commutation matrix. Therefore, the token-mixing block of the S-Mixer is

$$U = X + \text{Mat} \circ J_c^\top \circ \text{vec} \circ f_W \circ \text{Mat} \circ J_c \circ \text{vec} \circ \text{LN}(X) \tag{S.10}$$
$$= X + \text{Mat} \circ \text{PK-Layer}_W(\text{LN}(X); J_c, J_c^\top). \tag{S.11}$$

Similarly, the channel-mixing block of the S-Mixer is equal to

$$Y = U + \text{Mat} \circ \text{PK-Layer}_V(\text{LN}(U); I, I), \tag{S.12}$$

where $I$ is the identity matrix. Note that skip-connections gather $J_c$ and $J_c^\top$ in the same mixing block for compatibility in shapes of hidden units.

To get examples of PK family and to generalize Mixers, we implement the random permuted (RP) S-Mixer with skip-connections by replacing $J_c$ and $J_c^\top$ with i.i.d. random permutation matrices $J_1$ and $J_2$:

$$U = X + \text{Mat} \circ \text{PK-Layer}_W(\text{LN}(X); J_1, J_2). \tag{S.13}$$

We implement the random permutations by random shuffling of output $m = SC$ indexes of vectors. We freeze it during the training step. Note that we avoid using an $m \times m$ matrix representation of $J_x$ for memory efficiency. We implement the random permuted (RP) MLP-Mixer by the same way as the RP-S-Mixer.

### A.2.4  THE SKIP-CONNECTION IN THE FIRST BLOCK

The first token-mixing block has the input shape $(S_0, C)$ and the output shape $(S, C)$. However, we need to change $S$ with fixing $S_0$ in some experiments. To control the difference of $S_0$ and $S$, we set the first token-mixing block as follows:

$$U = \text{SkipLayer}(X) + \text{PK-Layer}_W(\text{LN}(X); J_1, J_2), \tag{S.14}$$

where the skip layer is given by

$$\text{SkipLayer}(X) = \text{LN}(\tilde{W}X), \tag{S.15}$$

where $\tilde{W}$ is a $S \times S_0$ weight matrix. For a fair comparison, we use the skip layer even if $S = S_0$ in the experiments that we sweep $S$. We use the same setting for the MLP-Mixer as for the S-Mixer.

### A.2.5  SPARSE-WEIGHT (SW) MLP

Let $0 < p \le 1$ and $m \in \mathbb{N}$. We implement each matrix of a static sparse weight FC block with the freezing rate $p$ as follows:

$$x \mapsto x + \phi((M \odot W)\,\text{LN}(x)), \quad x \in \mathbb{R}^m, \tag{S.16}$$

where $M$ is the mask matrix whose $m^2 p$ entries are randomly chosen and set to be one with a probability $p$ and the others are set to be zero with a probability $1 - p$. The mask matrix $M$ is initialized before training and it is frozen during training.

We also consider the SW-MLP consists of sparse weight MLP-blocks as follows:

$$x \mapsto x + \phi\left((M_2 \odot W_2)\phi((M_1 \odot W_2)\,\mathrm{LN}(x))\right), \quad x \in \mathbb{R}^m. \tag{S.17}$$

$W_1$ and $W_2$ are weight matrices with hidden features $\gamma m$, where $\gamma$ is an expansion factor. $M_1, M_2$ are mask matrices whose $\gamma m^2 p$ entries are randomly chosen and set to be one with a probability 1 and the others are set to be zero with a probability $1 - p$.

SW-MLP with $L$-blocks is composed in the order of the per-patch FC layer, vectorization, $L$ static sparse weight FC blocks (or MLP blocks), and the last classification FC layer.

## B ANALYSIS

### B.1 DERIVATION OF PROPOSITION 4.1

For $H = \phi(\phi(WX)V)$, by using $\mathrm{vec}(WXV) = (V^\top \otimes W)\mathrm{vec}(X)$, we have

$$\mathrm{vec}(H) = \phi((V^\top \otimes I_S)\mathrm{vec}(\phi(WX))) \tag{S.18}$$

$$= \phi((V^\top \otimes I_S)\phi((I_C \otimes W)x)). \tag{S.19}$$

Because $J_c^\top(A \otimes B)J_c = (B \otimes A)$ Magnus & Neudecker (2019) and any permutation matrix $J$ is commutative with the entry-wise activation function: $J\phi(x) = \phi(Jx)$, we obtain

$$\mathrm{vec}(H) = \phi(J_c^\top(I_S \otimes V^\top)\phi(J_c(I_C \otimes W)x)). \tag{S.20}$$

It may be informative that a similar transformation between the matrix and vector is used in a completely different context of deep learning, that is, the Kronecker-factored Approximate Curvature (K-FAC) computation for natural gradient descent Martens & Grosse (2015). K-FAC assumes layer-wise preconditioner given by the Kronecker product, that is, $(B \otimes A)^{-1}\mathrm{vec}(\nabla_W Loss(W))$ where $A$ and $B$ correspond to the Gram matrices of the forward and backward signals. This K-FAC gradient can be computed efficiently because it is reduced to a matrix computation of $A^{-1}\nabla_W Loss(W)(B^\top)^{-1}$. Therefore, the trick of formulating a large matrix-vector product for the product among relatively small matrices is common between K-FAC and the aforementioned effective expression.

### B.2 PRODUCT OF RANDOM PERMUTATIONS

The uniformly distributed random $m \times m$ permutation matrix is given by $J = J_g$ where $g$ is the uniformly distributed random variable on the permutation group $S_m$ of $m$ elements. Then the uniform distribution over $S_m$ is the Haar probability measure, which is translation invariant (see Folland (2013) for the detail), that is, $J = J_\sigma J_g$ is also uniformly distributed if $\sigma$ is $S_m$-valued uniformly distributed random variables and $g \in S_m$ is constant. Therefore, $J = J_\sigma J_\rho$ is a uniformly distributed random permutation matrix for independent and uniformly distributed random variables $\sigma$ and $\rho$ on $S_m$.

### B.3 REPRESENTATION AS A PK-FAMILY

By comparing (7) and (10), one can see that the block of the S-Mixer is

$$U = \text{PK-Layer}_W(X; I, J_c), \ \text{PK-Layer}_{V^\top}(U; I, J_c^\top) \tag{S.21}$$

For MLP-Mixer, the Token-MLP is

$$U = \text{PK-Layer}_{W_2}(\text{PK-Layer}_{W_1}(X; I, J_1); J_1^\top, J_c) \tag{S.22}$$

and Channel-MLP is

$$\text{PK-Layer}_{W_4^\top}(\text{PK-Layer}_{W_3^\top}(U; I, J_2); J_2^\top, J_c^\top) \tag{S.23}$$

for the arbitrary permutation matrices $J_1$ and $J_2$.

### B.4 DISCUSSION ON MONARCH MATRIX

Corollary 3.2 provides an interpretation of the Mixers: The use of FC layers with Monarch matrices has been experimentally demonstrated to be efficient in terms of memory and time. However, as $\sqrt{n}$ (i.e., the number of block matrices and the size of their widths) increases, it becomes more challenging to store them in memory despite the sparseness of the Monarch matrix. In Section B.4, we show the detail. The MLP-Mixer approximates the block diagonal matrix of the Monarch matrix by sharing the same block and allowing an efficient $\sqrt{n} \times \sqrt{n}$ matrix expression $H$ (i.e., approximating the MLP with the Monarch matrix by the PK family). Although we need to restrict the activation function in the intermediate layer to a linear function, it is interesting that the Mixers can implement Monarch matrices as a special case. If the S-Mixer includes a sequential part where the linear activation function is applied successively $L$ times, we can also implement the product of Monarch matrices under weight sharing, such as $\phi(M_L \cdots M_2 M_1 x)$.

## C EXPERIMENTAL SETTING

### C.1 FIGURE 1

We set the number of blocks of the MLP-Mixer, denoted by $L$, to 3. Both the Token-mixing block and the Channel-Mixing block are present $L$ times each, resulting in a total of 6 blocks when counted separately. We also set $\gamma = 2$.

For the comparison, the sparse-weight MLP replaces the components within the MLP-Mixer, namely $(I_C \otimes W_1), (I_C \otimes W_2)$ and $(W_3^\top \otimes I_S), (W_4^\top \otimes I_S)$, with the form $M \odot A$. In this context, there's no distinction between the token-mixing and the channel-mixing blocks, leading to a total of 6 blocks.

In Figure 1 (a), we compare the MLP-Mixer with $S = C = 64, 32$ and the SW-MLP. The sparsity of the SW-MLP is taken as $2^{-n}$ where $n$ ranges from 0 to 10. We set the patch size as 4. Each network is trained on CIFAR10 with a batch size of 128, for 600 epochs, a learning rate of 0.01, using auto-augmentation, AdamW optimizer, momentum set to 0.9, and cosine annealing. We utilize three different random seeds for each training.

Figure 1(b) shows the CKA (for a specific seed) of the MLP-Mixer with $S = C = 64$ and the MLP with a sparsity of 1/64, based on the results from Figure 1(a). However, the features targeted for CKA are taken from the layer just before the skip-connection in each block, and they are displayed on the axis in the order of proximity to the input, labeled as 1 through 6. Figure 1(c) similarly compares the dense MLP (i.e. sparsity= 1).

In Figure 1(d), we compare the MLP-Mixer and the MLP under the same training settings as in Figure 1(a). Now, here $\Omega = 2^{19}$ and $\gamma = 2$, with $C$ values being 4,8,12, 16, 20, 24, 28, 32, 36, 40, 44, 48, 64. The $S$ values are determined corresponding to $C$ while keeping $\Omega$ fixed, resulting in values: 173, 152, 136, 124, 113, 104, 96, 89, 83, 64. We utilized four different random seeds for each training.

### C.2 TABLE 1

We used single Tesla V100 GPU to compute the runtime. The training is by AdamW with the mini-batch size 128. We used 32-bit float. The runtime is averaged over 600 epochs. The averaged runtime is on four different random seeds.

### C.3 TABLE 2

For our experiments, we utilized Tesla V100 GPUs, accumulating approximately 300 GPU hours. The networks were trained on datasets, either CIFAR-10 or CIFAR-100. We set $L = 2$ and $\gamma = 4$. For the Mixer-SS/8 configuration, the parameters were set as $p = 8$, $S = 16$, and $C = 986$. In our specific approach, the parameters were defined as $p = 4$, $S = 64$, and $C = 487$. The training was conducted over 4000 epochs with the cosine-annealing. We employed the AdamW optimizer and incorporated auto-augmentation techniques. The chosen mini-batch size was 1024, and experiments were run with three distinct random seeds. To optimize results, we conducted a hyper-parameter search for the best initial learning rate from the set $\{0.04, 0.05, 0.06\}$. The rate that provided the highest accuracy, when averaged over the random seeds, was subsequently chosen for experiments.

Specifically, the learning rate 0.06 was selected for Mixer-SS/8 on CIFAR-100, while 0.04 was the preferred rate for all other scenarios.

## C.4 FIGURE 3

We utilized Tesla V100 GPUs and approximately 400 GPU hours for this experiment. We trained three types of MLPs; S-Mixer, RP S-Mixer, and SW-MLP architectures. All MLPs incorporated a per-patch FC layer as the first block, with a patch size of $P = 4$. The input token size was fixed at $S_0 = (32/P)^2 = 64$. We trained the models on the CIFAR-10, CIFAR-100, and STL-10 datasets, along with data augmentations such as random cropping and random horizontal flipping. The input images were resized to a size of $32 \times 32 \times 3$. We employed Nesterov SGD with a mini-batch size 128 and a momentum of 0.9 for training, running for 200 epochs. The initial learning rate was set to 0.02, and we used cosine annealing for learning rate scheduling. To ensure robustness, we conducted three trials for each configuration and reported the mean and standard deviation of the results. Unless otherwise specified, these settings were used throughout the whole study on CIFAR-10, CIFAR-100, and STL-10.

**(i) S-Mixer and RP S-Mixer** We conducted training experiments on the S-Mixer architecture with eight blocks. In order to explore various cases of integer pairs $(S, C)$ that approximately satisfy the equation

$$\Omega = \frac{CS^2 + SC^2}{2}. \tag{S.24}$$

The number of connections, denoted as $\Omega$, was fixed at $\Omega = 2^{18}, 2^{21}, 2^{27}$. For each value of $\Omega$, the pairs $(C, S)$ were chosen in a symmetric manner. It should be noted that if $(C, S) = (a, b)$ is a solution, then $(C, S) = (b, a)$ is also a solution. The selected pairs for each value of $\Omega$ are as follows:

- $\Omega = 2^{18}$: $(C, S) = (16, 173), (32, 113), (48, 83), (64, 64), (83, 48), (113, 32), (173, 16)$.
- $\Omega = 2^{21}$: $(C, S) = (16, 504), (32, 346), (64, 226), (128, 128), (226, 64), (346, 32), (504, 16)$.
- $\Omega = 2^{27}$: $(C, S) = (128, 1386), (256, 904), (512, 512), (904, 256), (1386, 128)$.

**(ii) SW-MLP.** For a fair comparison between the Mixers and SW-MLPs, we set the first layer of both models to the same per-patch FC structure. We trained SW-MLPs with eight blocks, where the hidden layers of these MLPs share a common $\Omega = 2^{18}$ connectivity pattern. Following the per-patch fully connected layer, the feature matrix is vectorized and processed as a standard MLP with masked sparse connectivity.

For each freezing rate $1 - p$, we determined the width $m$ of the hidden units using the equation:

$$\Omega = m^2 p, \quad m = \sqrt{\frac{\Omega}{p}}. \tag{S.25}$$

We set $1 - p = 0.1, 0.3, 0.5, 0.7, 0.9$, which correspond to $m = 540, 612, 724, 935, 1619$, respectively.

## C.5 FIGURE 4

We trained eight-block models for each datasets.

**CIFAR-10, 100, STL-10.** We utilized Tesla V100 GPUs and approximately 200 GPU hours for our experiments. We used a single GPU per each run. We set $\Omega = 2^{18}$ and used the same pairs $(S, C)$ satisfying equation S.24 as Sec. C.4.

We set the initial learning rate to $0.1$. On CIFAR-10, we trained models 200 epochs, 600-epochs on CIFAR-100, and we did five trials with random seeds. We trained models 2000-epochs on STL-10 with three random seeds.

**ImageNet-1k.** We utilized Tesla V100 GPUs and approximately 4000 GPU hours for our experiments. for training MLP-Mixer and RP MLP-Mixer on ImageNet-1k; we used a GPU cluster of 32 nodes of 4 GPUs per node for each run.

We set the expansion factor $\gamma = 4$ for both token-mixing MLP and channel-mixing MLP. We set $\Omega = 290217984 = (768^2 \cdot 196 + 768 \cdot 196^2)\gamma/2$ on a baseline $P = 16, (S, C) = (196, 768)$. We sweep $P = 7, 8, 14, 16, 28, 32$ and set $C = 3P^2$ and set $S$ so that it approximately satisfies the equation

$$\Omega = \frac{\gamma(CS^2 + SC^2)}{2}. \tag{S.26}$$

For each setting, we did three trials with random seeds.

Training on the ImageNet-1k is based on the timm library Wightman (2019). We used AdamW with an initial learning rate of $10^{-3}$ and 300 epochs. We set the mini-batch size to 4096 and used data-parallel training with a batch size of 32 in each GPU. We use the warm-up of with the warm-up learning rate $10^{-6}$ and the warm-up epoch 5. We used the cosine annealing of the learning rate with a minimum learning rate $10^{-5}$. We used the weight-decay 0.05. We applied the random erasing in images with a ratio of 0.25. We also applied the random auto-augmentation with a policy rand-m9-mstd0.5-inc1. We used the mix-up with $\alpha = 0.8$ and the cut-mix with $\alpha = 1.0$ by switching them in probability 0.5. We used the label smoothing with $\varepsilon = 0.1$.

### C.6 TABLE 2

We set $L = 8, p = 14, C = 588, S = 256$, and $\gamma = 4.57$. The other settings are the same as in Figure 4(d).

### C.7 FIGURE 5

We conducted our experiments using Tesla V100 GPUs, with a total of approximately 100 GPU hours utilized. For our experiments on CIFAR-10, we trained both MLP-Mixer and RP MLP-Mixer models. We set the initial learning rate to be 0.1.

**Left Figure:** In the left figure, we considered different values of $\Omega$, specifically $\Omega = 2^{19}, 2^{20}, 2^{21}, 2^{22}$. The output dimension of the per-patch FC layer was set to 64, 128, 256, 512, respectively. Additionally, we varied the value of $C$ as $C = 32, 64, 128, 256, 512, 1024$. The expansion factor, denoted as $\gamma$, was determined by the equation

$$\gamma = \frac{2\Omega}{CS^2 + SC^2}. \tag{S.27}$$

We plotted the runs where the training accuracy exceeded 0.8 after training.

**Right Figure:** In the right figure, we fixed $\Omega = 2^{21}$ and $m = 2^{12}$. We set the pairs of $C = 16, 32, 64, 83, 113, 173$ and $S = m/C$. The expansion factor was determined using Equation equation S.27. We performed three trials for each setting with random seeds.

### C.8 FIGURE 6

For our experiments in Figure 5, we utilized Tesla V100 GPUs, with approximately 70 GPU hours utilized. We trained both S-Mixer and RP S-Mixer models on CIFAR-10, CIFAR-100, and STL-10 datasets. We considered different numbers of blocks, specifically $L = 4, 8, 12, 16, 20$. The values of $S$ and $C$ were fixed at 128. Each configuration was evaluated using three trials with different random seeds.

## D SUPPLEMENTARY EXPERIMENTS

### D.1 TRAINABILITY OF HIGHLY SPARSE WEIGHTS

Golubeva et al. (2021) found that as the sparsity (width) increased to some extent, the generalization performance improved. They also reported that if the sparsity became too high, the generalization

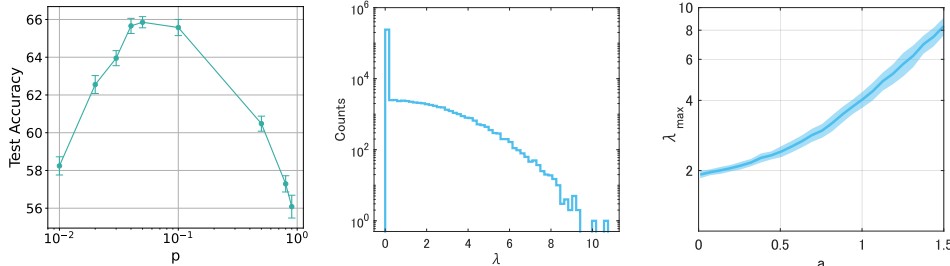

Figure S.1 : On the trainability of SW-MLP. (Left) Trainability decreases as the sparsity $1 - p$ becomes too high. We set $\gamma$ according to Golubeva et al. (2021) under fixed $\Omega = 2^{16}$. We performed five trials for each $p$ with random seeds. (Center) The singular value distribution of the sparse weight at random initialization. We set $a = 1.5$, $\Omega = 10^3$ and performed 50 trials. (Right) The largest eigenvalue monotonically increases as the sparsity increases.

performance slightly decreased. They discussed that this decrease was caused by the deterioration of trainability, that is, it became difficult for the gradient descent to decrease the loss function. In fact, we confirmed their decrease of the performance in SW-MLP as is shown in Figure S.1 (left). In contrast, we hardly observed such a decrease in performance for the Mixers. This seems rational because we can take an arbitrary small sparsity $1 - p$ for the SW-MLP while it is lower-bounded for the Mixers as is described in Section 4.

As a side note, we give here quantitative insight into the trainability from the perspective of the singular value of the weight matrix. Some previous work reported that the large singular values of weight matrices at random initialization cause the deterioration of trainability in deep neural networks Bjorck et al. (2018). Following this line of study, let us consider a random weight of SW-MLP. Set the width by $m = \Omega^{(1+a)/2}$ and the sparsity by $p = 1/\Omega^a$ with a constant $a > 0$ and take a large $\Omega$ limit. We use this scaling because our interest is in the case where the expected number of weights, i.e., $m^2 p = \Omega$, is independent of the scaling of $p$. We generate $Z = M \odot W$ where $W_{ij} \sim \mathcal{N}(0, 1)$ and $M$ is a static mask matrix whose entries are given by the Bernoulli distribution with probability $p$. The singular value of the weight matrix $Z$ is equivalent to the square root of the eigenvalue of $Q = ZZ^\top$. Because we are uninterested in a trivial scale factor, we scaled the matrix $Q$ as $Q/c$ where $c$ denotes the average over the diagonal entries of $Q$. This makes the trace of $Q$, that is, the summation (or average) of eigenvalues, a constant independent of $a$. We computed the eigenvalues of $Q$ and obtained the singular values of $Z$ denoted by $\lambda$.

As is shown in Figure S.1 (center), the spectrum of the singular values peaked around zero but widely spread up to its edge. Figure S.1 (right) demonstrates that the largest singular value becomes monotonically large for the increase of $a$. Because the larger singular value implies the lower trainability Bjorck et al. (2018), this is consistent with the empirical observation of Golubeva et al. (2021) and our Figure S.1 (left).

In contrast, the Mixers are unlikely to suffer from the large singular values as follows. Suppose S-Mixer with $S = C \gg 1$ for simplicity. Then, each layer of the effective MLP has $p = 1/C$ which corresponds to the scaling index $a = 1/3$ in SW-MLP. Actually, its singular value becomes further better than $a = 1/3$, because the weight matrices of the normal and RP Mixers are structured: Consider the singular values of $Z = J_2(I_C \otimes W)J_1$ with a $C \times C$ random Gaussian matrix $W$ and permutation matrices $(J_1, J_2)$. Then, the singular values of $Z$ are equivalent to those of $W$, excluding duplication. Therefore, the singular values of the Mixers are determined only by the dense weight matrix $W$. Define $Q = WW^\top$. Because the normalized matrix $Q/c$ obeys the Marchenko-Pastur law in the random matrix theory and its largest eigenvalue is given by 4 in the infinite $C$ limit Bai & Silverstein (2010). This means that the largest singular value of the normalized $W$ is 2 and corresponds to $a = 0$ of SW-MLP (i.e., dense MLP) with the infinite $\Omega$ limit in Figure S.1 (right). Thus, we can say that from the perspective of random weights, the trainability of the Mixers is expected to be better than that of SW-MLP.

We can also extend our analysis to the models incorporating the expansion factor $\gamma$. For SW-MLP with MLP blocks, the expected number of weights is given by $\Omega = \gamma p m^2$. We just need to replace $p$

in the S-Mixer case to $\gamma p$ and the monotonic increase of the largest singular value appears as well. For the MLP-Mixer, its normalized $W$ is a $\gamma C \times C$ matrix. According to the Marchenko-Pastur law for rectangular random matrices, as $C \to \infty$, the largest singular value approaches a constant value of $1 + \sqrt{\gamma}$. This corresponds to the singular value of $a = 0$ in the corresponding SW-MLP, and the result is similar as in the S-Mixer.

Note that the effective width of the mixing layers is sufficiently large but still has an upper bound (13). It satisfies

$$(\sqrt{1 + 8\Omega/\gamma} - 1)/2 \leq m \leq (\Omega/\gamma)^{2/3}, \tag{S.28}$$

where the equality of the lower bound holds for $S = 1$ or $C = 1$. In contrast, for SW-MLP, we have no upper bound and only the lower bound $\sqrt{\Omega} \leq m$, where this equality holds for a dense layer. We can consider an arbitrarily small $p$ and a large $m$ for a fixed $\Omega$ if we neglect the issue of memory (Golubeva et al., 2021). Golubeva et al. (2021) reported that extremely small $p$ can cause a decrease in test accuracy owing to the deterioration of trainability. We observed a similar deterioration for the SW-MLP, as shown in this section, but not for the Mixers. This is expected because $m$ is upper-bounded in the Mixers and the trainability is less damaged than that of the SW-MLP with high sparsity.

## D.2 MLP-MIXER WITH INCREASING WIDTH

Figure S.2 shows the test accuracy improves as the width increases on the models SW-MLP and normal and RP MLP-Mixer even if the expansion factor $\gamma$ is incorporated in the models. We set the expansion factor $\gamma = 4$. We set the initial learning rate to be 0.1. For normal and RP MLP-Mixer, we set $C = 16, 32, 48, 64, 83, 113, 173$ and determined $S$ by combinations of $C$ and $\Omega = 2^{18}, 2^{20}$. For SW-MLP, we set $p = 0.1, 0.3, 0.5, 0.7, 0.9$ and set the width by $m = \sqrt{\Omega/\gamma p}$.

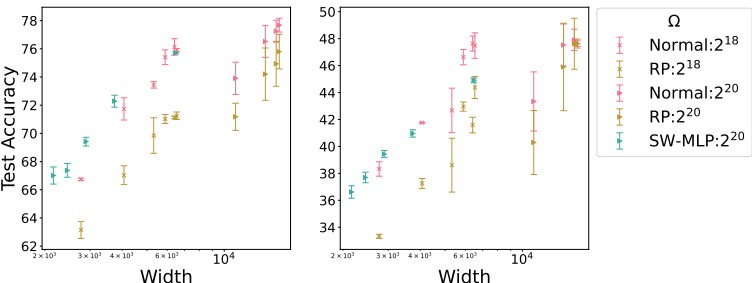

Figure S.2 : Test accuracy improves as the effective width increases. MLP-Mixer, RP-MLP-Mixer, and SW-MLP with $\gamma = 4$ on (Left) CIFAR-10, (Right) CIFAR-100.

## D.3 REPLACEMENT OF $J_c$ IN SPECIFIC LAYERS

Figure S.3 provides more detailed insight into the case where the depth is limited and RP Mixers perform worse than normal Mixers. We investigated special S-Mixers whose $l$-th block was replaced with its RP counterpart while the other layers remained the same. Interestingly, when the accuracy deterioration apparently appears (e.g., cases of CIFAR-10 and STL-10 in Figure 6), this deterioration is attributed to the first block. This seems rational because the neighboring image patches are likely to be correlated, which makes the input units to the first token mixing correlated. Although the usual mixing weights can reflect such neighboring structures, RP Mixers randomly choose tokens and may lose the neighboring structure specific to the images. However, as the depth increases, the token mixing layers can merge all the tokens, which is consistent with the increase in the accuracy of the RP Mixers, as confirmed in Figure 6. Thus, we conclude that the RP and normal mixing layers have almost the same inductive bias, especially, in the upstream layers.

We utilized Tesla V100 GPUs and approximately 10 GPU hours for our experiments. Consider the S-Mixer architecture consisting of four blocks and $S = C = 64$. In this study, we trained a modified version of the S-Mixer architecture by replacing one of the four blocks with a block that incorporates random permutation. The training was conducted on CIFAR-10, CIFAR-100, and STL-10 datasets.

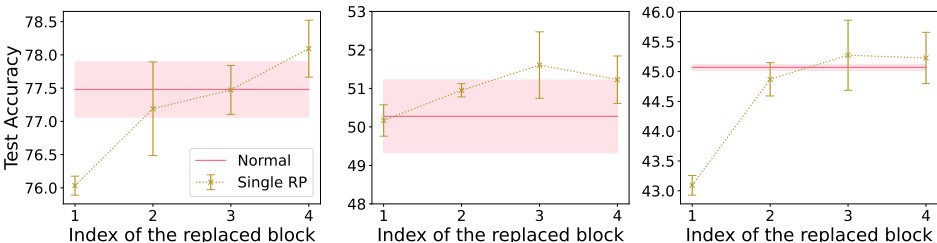

Figure S.3 : Replacing a single block of the normal Mixer with a corresponding RP block clarifies that the upstream layers are functionally commutative with the RP block. (Left) CIFAR-10, (Center) CIFAR-100, (Right) STL-10. We set $C = S = 128$.

The optimizer and training settings used in this experiment were consistent with those described in Section C.8.

## D.4 REMARK ON INPUT PATCH SIZE

In this study, we focused on changing the size of the mixing layers and fixed the input token and channel size $(S_0, C_0)$. In other words, the patch size $P$, satisfying $C_0 = 3P^2$, is fixed. While we observe that our experimental results hold regardless of the patch size, one naive question is whether there is any optimal patch size for achieving the highest accuracy. Although this is beyond the scope of this study, we show the performance depending on $C_0$ in Figure S.4 as a side note. The number of mixing layers is fixed at $C = S = 64$. We observed that the optimal $C_0$ depended on data; $C_0 = 48$ ($S_0 = 64$) for CIFAR-10 and 100, and $C_0 = 108$ ($S_0 = 225$) for STL-10. Note that the dimension of an input image is $S_0 C_0 = 3,072$ for CIFAR datasets and $24,300$ for STL-10. It would be rational that the optimal patch size depends on the detailed information of data.

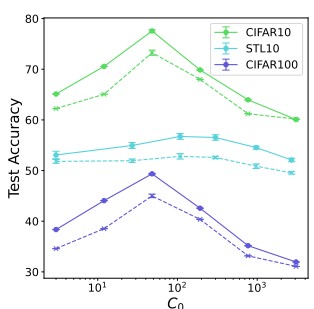

Figure S.4 : Dependence on $C_0$

We utilized Tesla V100 GPUs and approximately 30 GPU hours for our experiments. We conducted training experiments on CIFAR-10, CIFAR-100, and STL-10 datasets using both S-Mixer and RP S-Mixer architectures with $S = C = 64$, along with four blocks. For optimization, we set the initial learning rate to be 0.1.

For CIFAR-10 and CIFAR-100, we trained the models for 200 epochs and evaluated them with different patch sizes ($P = 1, 2, 4, 8, 16, 32$). We performed three trials with different random seeds for each setting. On the STL-10 dataset, we resized the images to $90 \times 90 \times 3$ and trained the models for 400 epochs. We varied the patch size ($P = 1, 3, 6, 10, 18, 30$) and performed five trials with different random seeds for each setting.

## D.5 RP CAN PREVENT THE OVERFITTING

In Figure S.5 , we explored several values of $C$ fixed $\Omega$, the normal model shows overfitting of more than twice the magnitude compared to the RP model, especially $C$ is the largest one in the exploring range. In this case, $S$ takes the smallest value among the explored values. This suggests that RP has a regularization effect beyond the token-mixing side and affects the channel-mixing side, particularly when $C$ is large.

To mitigate overfitting, additional augmentation (auto-augmentation, based on Cubuk et al. (2019)) was applied to the dataset, and the model was switched from S-Mixer to MLP-Mixer ($\gamma = 4$) due to the observed slower decrease in training loss for S-Mixer in Figure S.6 . RP S-Mixer outperformed S-Mixer in terms of test loss for $C = 173$, indicating that RP still provides overfitting prevention even with relatively strong data augmentations.

Figure S.6 (right) illustrates that RP did not reach the relatively small training loss as the normal model. To address this, SGD was replaced with AdamW as the optimizer, with a reduced initial learning rate (lr = 0.01) due to the instability observed with lr = 0.1 in Figure S.7 . This resulted

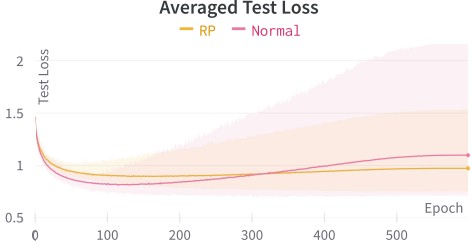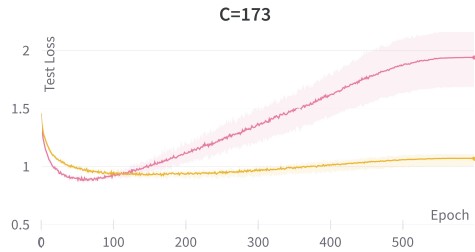

Figure S.5 : (Left) The average test loss curves are shown for $C = 16, 32, 64, 114, 173$ and five trials with different random seeds. The models used in this experiment were Normal and RP S-Mixers trained on CIFAR-10 with $L = 8$ for 600 epochs. The initial learning rate was set to 0.1. (Right) The test loss curve for $C = 173$ represents the worst case of overfitting. The shaded area in both figures represents the range between the maximum and minimum values.

in reduced overfitting in the $C > S$ region, and RP performed exceptionally well compared to the normal model for $C = S = 64$. In Figure S.7 , neither the normal nor RP models exhibited a significant increase in test loss for $C = S$. However, while the normal model's test loss plateaued, the RP model continued to decrease its test loss, eventually surpassing the normal model in terms of test accuracy. This highlights the potential of RP to outperform the normal model with a choice of optimization such as AdamW.

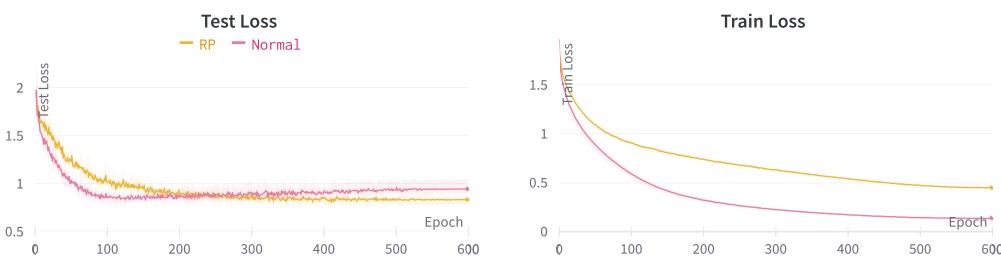

Figure S.6 : (Left) Average of test loss curves. (Right) Average train loss with $C = 173$. In both figures, the area shows max and min values. We trained normal and RP MLP-Mixer with $\gamma = 4$ with an initial learning rate of 0.1 and a mini-batch size of 256 for 600 epochs. The results are average of five trials. The shaded area in the figure represents the range of values between the maximum and minimum values.

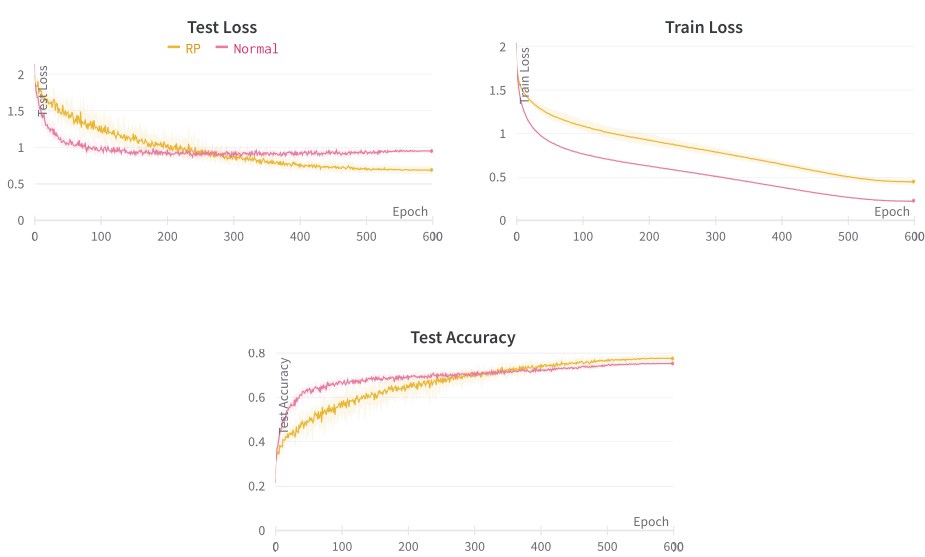

Figure S.7 : Results of training with AdamW and auto-augmentation with $S = C = 64$. The RP MLP-Mixer exceeded the results of the normal one in test loss and test accuracy. (Left) Average of test loss curves. (Right) Average train loss. (Lower) Test Accuracy curves. We set the initial learning rate to be 0.01 with a mini-batch size of 256 and 600 epochs. In all figures, the results are average of five trials and the area shows the range between the maximum and minimum values.

