# OpenReview forum: "Understanding MLP-Mixer as a wide and sparse MLP"
_ICLR.cc/2024/Conference — Submitted to ICLR 2024_

### Official Review · Reviewer_xS5y · 2023-10-31

**Soundness:** 2 fair
**Presentation:** 3 good
**Contribution:** 2 fair
**Rating:** 6
**Confidence:** 2

**Summary:**

This paper shows an effective expression of MLP-Mixer as an MLP by vectorizing the mixing layers. This paper clarifies that the mixing layer of the Mixer has an effective expression as a wider MLP whose weights are sparse and represented by the Kronecker product. It is also regarded as an approximation of Monarch matrices. This paper also introduced the RP-Mixer, a more memory-efficient alternative to the unstructured sparse-weight MLP.

**Strengths:**

1. This paper shows an effective expression of MLP-Mixer as an MLP by vectorizing the mixing layers
2. This paper finds similar tendencies between the mixer and unstructured sparse-weight MLP in hidden features and the performance under widening with a fixed number of connections.
3. This paper characterizes the Mixers as a special example of a general class: Permuted-Kronecker (PK) family

**Weaknesses:**

1.	The authors claim that wider Mixer is a an effective expression of MLP-Mixer and RP-Mixer is a more memory-efficient alternative to unstructured sparse-weight MLP. It would be beneficial to apply wider Mixer/RP-Mixer to more state-of-the-art frameworks [1,2,3,4,5], especially considering the more efficient MLP-Mixer approach proposed in [6].

2.	The paper lacks a thorough comparison with existing methods in terms of both memory efficiency and performance. It would be valuable to include such comparisons to demonstrate the superiority of wider- Mixer/RP-Mixer.


[1] Morphmlp: A self-attention free, mlp-like backbone for image and video. ECCV 2022.

[2] Sparse mlp for image recognition: Is self-attention really necessary? AAAI, 2022.

[3] As-mlp: An axial shifted mlp architecture for vision. ICLR, 2022.

[4] CycleMLP: A MLP-like architecture for dense prediction. ICLR, 2022.

[5] Active Token Mixer. AAAI, 2023.

[6] Adaptive Frequency Filters As Efficient Global Token Mixers. ICCV, 2023.

As a non-theoretical machine learning researcher, I am only able to give a recommendation based on the novelty of the idea and the empirical parts, and would give a borderline accept with low confidence according to the listed strengths and weaknesses.

**Questions:**

1. The paper lacks a comprehensive comparison with state-of-the-art (SOTA) methods. It is important to compare the proposed MLP-Mixer with other SOTA methods in terms of performance, memory efficiency, and any other relevant metrics.

2. Additionally, it would be beneficial to conduct a detailed analysis of existing MLP-Mixer variants or related methods. Exploring the strengths and weaknesses of these approaches would provide a more comprehensive understanding of the proposed method and its contributions to the field.

3. The experimental section should include a thorough evaluation of the proposed method's performance compared to other SOTA methods. This evaluation should consider multiple benchmark datasets and provide statistical analysis to support the claims of superiority.

---

> ### Author Response · Authors · 2023-11-21
> **Response to Reviewer xS5y:   Our Standing Position**
>
> Thank you for your helpful suggestion. We especially appreciate your providing numerous references. Our response to your comments is as follows.
>
> > The authors claim that wider Mixer is a an effective expression of MLP-Mixer and RP-Mixer is a more memory-efficient alternative to unstructured sparse-weight MLP. It would be beneficial to apply wider Mixer/RP-Mixer to more state-of-the-art frameworks [1,2,3,4,5], especially considering the more efficient MLP-Mixer approach proposed in [6]
>
> and,
>
> >Q1, Q2.
>
> As Reviewer suggests, research comparing MLP-Mixer with other Mixer variants is making significant contributions while being abundant. However, what we want to emphasize is that our objective is to elucidate **why the MLP-Mixer performs better than a naive MLP**. Therefore, our comparison is primarily with a naive MLP.
>
> The following explanatory diagrams and comparison tables **introduce the RP-Mixer to bridge MLP and MLP-Mixer**, not to propose a superior MLP-block structure design to MLP-Mixer. As shown in the comparison table, studies like [1,2,3,4,5,6] focus on innovating the structure within blocks for inter/intra-token mixing. In contrast, the RP-Mixer, which flattens tensors and mixes them randomly, is closer to a naive  MLP.
>
> -----
>
> MLP  — ( RP-Mixer)  —  MLP-Mixer   —   [1,2,3,4,5,6]
> -----
> Figure. Relationship between models.
>
> | Models  |    where  Blocks operate  on | Destroy/Improve structure of MLP-Mixer  |
> |--------|--------|--------|
> | MLP |              **whole**                |   **Destroy** |
> |  RP-Mixer  |     **whole** |     **Destroy** ( Flatten → Randomly Permute → Reshape)  |
> |MLP-MIxer |     inter/intra-tokens|  - |
> |MorphMLP[1] |  inter/intra-tokens| Improve (Separate H and W)|
> |sMLPNet[2] | inter/intra-tokens| Improve (Separate H and W)|
> |As-MLP[3] | inter/intra-tokens| Improve (Axial Shift)|
> |CycleMLP[4] |inter/intra-tokens| Improve (Axial Shift)|
> |ATM[5]  |  inter/intra-tokens |  Improve (Message Passing)|
> |AFF[6] |  inter/intra-tokens|  Improve (Adaptive Filter)|
>
> Table.  Comparison of models on their structure.
>
> The references [1-6] by Reviewer  experimentally reveal how modifying the mixing block can improve accuracy over the MLP Mixer. However, the fundamental question of **why a mixing block is superior to the dense layer of a naive MLP** remains a mystery. Understanding the intrinsic properties of these blocks will likely guide future improvements to Mixers, similar to references [1-6]. For instance, the equality $S=C$ is a quantitative suggestion. While Mixers with such a structure present an interesting problem, as mentioned earlier, our study's scope is to clarify the differences with a naive MLP, so we leave this for subsequent work in the future.
>
> Regarding the comparison with MLP:
>
> - In terms of efficiency, the RP-Mixer is about 10^2 to 10^3 times better than WS-MLP (Table 1 in the revised manuscript).
> - The state-of-the-art accuracy for MLP, represented by Nesybur's beta-laaso, is 85% (CIFAR10), whereas our proposed performance is 87.98% (CIFAR10), (Table 2).
> - While Scaling MLPs[Bachmann2023] achieves 56% on ImageNet with a naive MLP after various modifications, the RP-Mixer achieves 70% (ImageNet) as shown in Figure 6.
>
> [[Bachmann2023] Gregor Bachmann, et.al., Scaling MLPs: A Tale of Inductive Bias](https://neurips.cc/virtual/2023/poster/71680), NeurIPS2023.
> >The experimental section should include a thorough evaluation of the proposed method's performance compared to other SOTA method
>
> To achieve our purpose, a thorough evaluation means comparing MLP with MLP/RP-Mixer across a variety of parameters, widths, and numbers of layers. We have conducted performance comparison experiments using 5000 to 10000 GPU hours, which can be considered a thorough evaluation in this context.
>
> >comprehensive comparison with state-of-the-art (SOTA) methods
>
> We agree that it is one of the important contributions to improve the SOTA method. However, we expect that giving a unified perspective over existing fundamental methods (in our case, naive MLP and MLP-Mixer, that is, basic layers without additional structures mentioned in [1-6])  is also helpful in creating a solid starting point for architecture development in the future.

---

### Official Review · Reviewer_9wTw · 2023-10-31

**Soundness:** 3 good
**Presentation:** 3 good
**Contribution:** 3 good
**Rating:** 6
**Confidence:** 3

**Summary:**

This paper analyzes the MLP-Mixer model, and shows that the model can be restated as a wide MLP with sparse weights, and investigates some generalizations/variations of the model.
They show that these proposed variants perform well on CIFAR, STL and ImageNet.

**Strengths:**

Originality: While there is a large literature on the analysis of sparsity in neural nets, this specific type of analysis seems novel to me.
Quality: the analysis looks sound, the experiments are well done.
Clarity: I was able to follow along the paper.
Significance: The analysis done here might be applicable to a larger variety of models. Still, I think this might be the weakest part of the paper (see next field).

**Weaknesses:**

* The MLP-Mixer is a fairly nieche model that hasn't seen much adaption neither in practice nor as a vessel for theoretical analysis. So I'm under the impression that this work will not be immediately useful to a broader audience. However, the tools of analysis used and the results obtained might be useful for future research in related areas. The authors themselves mention such potential areas on the Conclusion.

* The authors do not mention running time for their variations, it would be nice if they would state both FLOPS and wallclock times.

**Questions:**

1) My main question to the authors would be if they have any immediate applications of their findings? I agree that in theory this demonstrates that you could use this to structure weights that can achieve a very large effective width. The MLP Mixer's MLP fulfills a very specific task (interlacing inter/intra-token processing), I'm unsure how to extrapolate that to other architectures.

2) What is the running time of their models compared to the original MLP Mixer formulation?

---

> ### Author Response · Authors · 2023-11-20
> **Response to Reviewer 9wTw: Application of Findings and Computational Requirements**
>
> Thank you for your helpful suggestions.  We updated the manuscript based on that.
>
> > My main question to the authors would be if they have any immediate applications of their findings.
>
> As (Golubeva, Neyshabur, and Gur-Ari (2021)) suggests, **sparsity and width are two critical factors** determining the performance of DNNs, and they have significantly contributed to the development of architecture and algorithms. We have discovered that MLPmixer is a model that efficiently and memory-friendly balances both these aspects.
>
> Therefore, it has the potential for much adaptation in practice and can serve as a vessel for theoretical analysis. For instance,
> (1)**replacing the MLP block with a Mixer-block** and considering a dual-structured Mixer-block as investigated in the research of **Monarch matrices**  [1-a] seems practically useful,
> (2) and it can be utilized for **verifying generalization bounds**[1-b, 1-c] based on width and sparsity, by replacing MLPs with MLP-Mixers. Because of computational efficiency, we can evaluate them on large datasets with practical performance.
>
> (1-a)[Monarch: Expressive Structured Matrices for Efficient and Accurate Training](https://proceedings.mlr.press/v162/dao22a/dao22a.pdf), ICML2022
>
> (1-b)[Norm-based Generalization Bounds for Sparse Neural Networks](https://openreview.net/forum?id=COPzNA10hZ), NeurIPS2023
>
> (1-c)[Generalization Bounds of Stochastic Gradient Descent for Wide and Deep Neural Networks](https://proceedings.neurips.cc/paper/2019/file/cf9dc5e4e194fc21f397b4cac9cc3ae9-Paper.pdf), NeurIPS2019
>
> > What is the running time of their models compared to the original MLP Mixer formulation?
>
> We have added not only the running time but also memory requirements and FLOPs to Table 1 in Section 4.2. As expected, **MLP-Mixer and RP-Mixer were comparable in running time, but SW-MLP was slower than them.** Furthermore, **RP-Mixer required $10^2$ ~   $10^3$ times less memory and FLOPs compared to  SW-MLP.**
>
> In conclusion, we hope our responses have successfully clarified any uncertainties you had, and that you see the potential in our research for future developments. Your positive reassessment would be greatly appreciated and encouraging for us.

---

> > ### Comment · Reviewer_9wTw · 2023-11-22
> >
> > Thank you for adressing my questions, it helps me better understand where you see its potential future applications. I think that overall the work is acceptable, but not outstanding -- while I personally would argue that it meets the bar for publication at ICLR, I think integrating some of the proposed future applications to show the value of the findings would make it much more convincing. I thus stand by my original rating.

---

> > > ### Author Response · Authors · 2023-11-22
> > > **Thank you for your comment!**
> > >
> > > Thank you for your feedback and for acknowledging that our response has clarified the potential applications of our work. We appreciate your view on its acceptability for ICLR. Based on your suggestion, we will explore integrating some of the proposed applications to enhance the paper's impact and make our findings more compelling. Thank you again for your constructive comments!

---

### Official Review · Reviewer_b7KH · 2023-11-02

**Soundness:** 4 excellent
**Presentation:** 3 good
**Contribution:** 2 fair
**Rating:** 5
**Confidence:** 3

**Summary:**

This paper made theoretical and practical study to understand the good performance of the MLP-Mixer architecture. More specifically, this paper shows that MLP-Mixer behaves similarly to a wide MLP with sparse weights. The wide and sparse MLP variant achieves comparable results compared to the original MLP-Mixer on CIFAR-10 and ImageNet-1k.

**Strengths:**

The idea of understanding MLP-Mixer as a wide and sparse MLP is original. This paper presents an analysis of the MLP-Mixer method, provides a good explanation to connect MLP-Mixer with the Kronecker product and shows that the model behaves as a wide MLP with sparse weights. It is a novel explanation to attribute the success of the Mixer architecture to the effective width of a sparse MLP. Experimental results show that the wide and sparse MLPs could achieve comparable results as the MLP-Mixer architecture.

**Weaknesses:**

1. Performance comparison of the MLP-Mixer with Wide MLP and RP-Mixer is missing. It would be nice to add the inference speed comparison and memory consumption comparison among the three methods (MLP-Mixer, Wide MLP, RP-Mixer).
2. The absolute results are a bit low on both CIFAR (84.1% baseline for Mixer) and ImageNet-1k (76.4% baseline for Mixer). It makes the improvements less convincing as 0.3 percent boost on ImageNet could easily be caused by many different reasons (augmentation, hyper-parameters, patch sizes, training durations, test-time crops, etc.) This may limit the impact of this paper.
3. The takeaways from this paper is a bit unclear, especially on how to properly make use of the new insights in this paper to either improve the quality or improve the performance of the existing architectures (i.e. MLP-Mixer).

**Questions:**

See above comments.

---

> ### Author Response · Authors · 2023-11-20
> **Response to Reviewer b7KH:  Performance Comparison and Takeaways from This Paper**
>
> Thank you for your helpful suggestions.  Based on your comments, we have updated the manuscripts.
>
> > Performance comparison of the MLP-Mixer with Wide MLP and RP-Mixer is missing. It would be nice to add the inference speed comparison and memory consumption comparison among the three methods (MLP-Mixer, Wide MLP, RP-Mixer).
>
> Thank you for your suggestion.  We added  Table 1 to the performance comparison in Section 4.2 in the revised version.   Table 1 shows that RP-Mixer is much superior to the naive wide MLP in memory requirements,  FLOPs, and inference speed.  As we expected, MLP-Mixer is comparable to RP-Mixer and they are much better than Wide MLP.
>
> > The takeaways from this paper is a bit unclear, especially on how to properly make use of the new insights in this paper to either improve the quality or improve the performance of the existing architectures (i.e. MLP-Mixer).
>
> As the reviewer mentioned, making improvements that significantly increase accuracy would indeed be a considerable contribution to research. However, since we are still at a stage where the reasons for the high performance of MLP-Mixer are not fully understood, providing an understanding of the mechanism where sparsity and width are crucial would be our sufficient contribution and takeaway. History has shown that contributing to the internal understanding of existing models, even when it doesn't directly lead to accuracy improvements, is considered a contribution within the community: for instance,
>
> **[Understanding Batch Normalization](https://proceedings.neurips.cc/paper_files/paper/2018/file/36072923bfc3cf47745d704feb489480-Paper.pdf)  NeurIPS 2018.**
>
> **[On the relationship between self-attention and convolutional layers](https://arxiv.org/abs/1911.03584) ICLR2020.**
>
> Our focus is not primarily on improving the existing architecture of MLP-Mixer, but rather on highlighting that using MLP-Mixer is more efficient than using a naive MLP  as the following (1), (2) and (3).
>
>
> (1)Replacing fully connected layers of existing models with Mixer blocks** could lead to performance improvements due to increased effective width. As discussed in Section 3.3, this relates to the **memory-efficient matrix representation** of MLPs with Monarch matrices, which is an interesting direction. For example,  in the MLP-Mixer,  if we replace each MLP block itself with a Mixer block then we have **dual structured Mixer blocks, which enlarges the effective width and can improve performance**.    However, proposing a higher-performance architecture would be too much content for a single paper and is left as an open problem for subsequent work.
>
> (2) As shown in the Appendix, Random permutations prevent overfitting and can be used with the dual Mixer block when we want to reduce undesirable implicit bias by usual Mixing blocks.
>
> (3)MLP is the most fundamental model and has been the subject of many theoretical studies[*]. However, training MLPs to a **practical level** on **large-scale datasets** is difficult from the perspectives of memory and computational cost, making it challenging to validate theories on large datasets.
>
> Through this research, we have discovered that the computationally efficient MLP mixer is similar to wide MLPs. This revelation **opens up the opportunity to experimentally validate previous theoretical studies on MLPs using an equivalent MLP-Mixer on large datasets!**
>
>  ref: * Examples of theories and possible applications.
> -   (3.1) MLP is known to fall into the NTK regime at infinite width, as discussed in [Jacot 2018]. **By using MLP-Mixer instead,  we can observe the order of finite width where feature learning occurs relative to the number of samples.**
> - (3.2) MLPs follow scaling laws if we prepare a  large dataset [Bachmann2023]. However, their performance was low (about acc. 50 % on ImageNet).  **Then by using the similarity of MLP-Mixer and MLP, we can observe how sparseness and wideness affect the scaling laws.**
>
> [[Jacot2018] Arthur, Jacot, et.al, Neural Tangent Kernel: Convergence and Generalization in Neural Networks](https://proceedings.neurips.cc/paper_files/paper/2018/file/5a4be1fa34e62bb8a6ec6b91d2462f5a-Paper.pdf), NeurIPS2018.
> [[Bachmann2023] Gregor Bachmann, et.al., Scaling MLPs: A Tale of Inductive Bias](https://neurips.cc/virtual/2023/poster/71680), NeurIPS2023.
>
>
>
>
> We hope that you can expect that our work will work as a concrete starting point for why the MLP-Mixer achieves higher performance compared to a standard MLP.
> We hope our explanations have clarified the issues you pointed out.   If so, we would be grateful for an adjustment in the score.

---

> > ### Author Response · Authors · 2023-11-22
> > **Further comments before author response period end ?**
> >
> > Thank you once more for dedicating your time to review our paper. As the discussion period nears its conclusion, we wish to ensure that our responses have adequately addressed your primary concerns. Please feel free to share any additional comments or queries; we are eager to respond to them.

---

> ### Comment · Reviewer_b7KH · 2023-12-03
>
> Thank you for adding Table 1 for the performance comparison, this has answered my question. I have read the authors' response on the papers' contributions, I do understand the impact of this paper to connect MLP-Mixer with wide and sparse MLP. It would make the paper more convincing by showing concrete quality or performance improvements. I thus keep my original rating.

---

### Official Review · Reviewer_xf5f · 2023-11-07

**Soundness:** 2 fair
**Presentation:** 3 good
**Contribution:** 2 fair
**Rating:** 6
**Confidence:** 3

**Summary:**

This paper explores the potential of MLP-Mixer as a wide and sparse MLP. The author demonstrates through the use of Kronecker product that the mixing layer of the Mixer has an effective representation as a wider MLP, which has sparse weights, and regards it as an approximation of the Monarch matrix. Additionally, the authors also introduce a more memory-efficient RP-Mixer to verify the similarity in much wider cases.

**Strengths:**

1. The author has conducted extensive parameter analysis to validate that the mixing layer of both the MLP-Mixer and the RP Mixer effectively represents a wider MLP.
2. The author offers a new analytical perspective to elucidate the effectiveness of the MLP-Mixer.

**Weaknesses:**

1. The paper falls short in terms of the selection of networks for comparison, thereby resulting in a lack of theoretical support.
2. There is a lack of experimental evidence to support the memory efficiency and lightweight structure of the RP-Mixer.
3. According to (Magnus and Neudecker, 2019), there appear to be slight mistakes in the theoretical proof section. For instance, formula $J_c^{\top}\left(I_S \otimes V\right) J_c=V^{\top} \otimes I_S$ should actually be $J_c^{\top}\left(I_S \otimes V\right) J_c=V \otimes I_S$.
Ref:
Jan R Magnus and Heinz Neudecker. Matrix differential calculus with applications in statistics and econometrics. John Wiley & Sons, 2019.

**Questions:**

1. The motivation for this article's research appears to be similar to that of (Golubeva and Neyshabur, 2021). Could you clarify how the author's approach to analyzing network width and sparsity differs from that in Article A?
2. In the contribution, why is the RP-Mixer referred to as a computationally demanding yet lightly-structured alternative? This statement seems contradictory.
Ref:
Anna Golubeva, Behnam Neyshabur, and Guy Gur-Ari. Are wider nets better given the same number of parameters? In International Conference on Learning Representations, 2021.

---

> ### Author Response · Authors · 2023-11-18
> **Response to Reviewer xf5f:  Motivation of this article and computational requirements**
>
> We appreciate the time and effort you put into reviewing our manuscript. Your helpful suggestions have greatly improved the quality of our paper.   Our point-to-point responses to your comments are given below.
>
> > The paper falls short in terms of the selection of networks for comparison, thereby resulting in a lack of theoretical support.
>
> Thank you for your feedback. We acknowledge the concern regarding the selection of networks for comparison and the perceived lack of theoretical support.  We want to emphasize that our goal is to understand a single model, the MLP-Mixer. To achieve this, we have conducted exhaustive experiments with various parameters (about 10 selections), widths (about 50 selections), and numbers of layers (about 10 selections). In this context, we believe that the selection of networks for comparison is adequate for our purpose.
>
> Additionally,  we note that these experiments required over a total of 5000 GPU hours.
>
> >There is a lack of experimental evidence to support the memory efficiency and lightweight structure of the RP-Mixer.
>
> Thank you for your suggestions. We have added Table 1 in Section 4.2 which shows the experimental evidence of the efficiency of RP-Mixer in memory requirements, FLOPs, and runtime. This successfully verifies our claim that RP-Mxier is comparable to MLP-Mixer, and they are much better than  SW-MLP!
>
> >formula   J_c^\top (I_S \otimes V)J_c = V^\top \otimes I_S should actually be  V \otimes I_S.
>
> Thank you very much for catching this confusing typo. We have corrected the omission of the transpose symbol in the formula.
>
>
> > The motivation for this article's research appears to be similar to that of (Golubeva and Neyshabur, 2021). Could you clarify how the author's approach to analyzing network width and sparsity differs from that in Article A?
>
> To avoid potential misunderstanding between the reviewer and us, let us emphasize that our motivation or starting point of the work is totally different from Golubeva et al 2021.
> Our motivation is to understand the mechanism of MLP-Mixer and why it works well, and it revealed that the sparsity structure of MLP-Mixer consists of the Kronecker product with the commutation matrix. In contrast, the motivation of Golubeva et al 2021 is to find the importance of sparseness in wide neural networks (in more detail, usual forward neural networks; naive MLP, and CNNs).
> The most interesting finding that we raised is that the seemingly different work on the sparseness by Golubeva et al 2021 has a hidden connection to the MLP-Mixer (through the Kronecker product expression). In other words, what we showed in the current work is that we can apply the same approach as in  Golubeva et al 2021 to the MLP-Mixer which is a “seemingly dense” network in the matrix expression but sparse in the vector expression.
>
> >  In the contribution, why is the RP-Mixer referred to as a computationally demanding yet lightly-structured alternative?
>
> Thank you for catching this typo. I have corrected the term “demanding” to be ”undemanding” in the section.
>
>
>
> We hope our responses have cleared up the uncertainties you identified, we would be grateful for an adjustment in the evaluation score.

---

> > ### Author Response · Authors · 2023-11-22
> > **Further comments before author response period end ?**
> >
> > Thank you once more for dedicating your time to review our paper. As the discussion period nears its conclusion, we wish to ensure that our responses have adequately addressed your primary concerns. Please feel free to share any additional comments or queries; we are eager to respond to them.

---

### Official Review · Reviewer_XJtr · 2023-11-09

**Soundness:** 3 good
**Presentation:** 2 fair
**Contribution:** 2 fair
**Rating:** 6
**Confidence:** 3

**Summary:**

The authors have shown that MLP mixers are essentially equivalent to a much wider MLP layer, which has structured sparsity:

1. The authors demonstrated a way to rewrite the formulation of MLP mixer into a standard MLP, where the input to the MLP is the vectored feature matrix and the wide MLP weights are constructed from the Kronecker product.
2. The authors empirically showed that the representations obtained from the equivalent wide MLP and MLP-Mixer are similar.
3. The authors also showed that the wide MLP resembles the Monarch matrix.
4. The authors also tried reducing the "structureness" of the equivalent wide MLP's sparsity, by introducing a permutation matrix.
5. Finally, the authors argued that an MLP mixer is a way to achieve wide MLP without the computation cost of the wide MLP, and increasing the effective width indeed improves the performance.

**Strengths:**

1. The derivation and observation seems to be solid.
2. I believe this is the first time the equivalence between MLP mixer and wide MLP has been formalized.

**Weaknesses:**

1. The paper is not very easy to follow. For one, a lot of notations are not defined, which require the reader to find out from the original MLP mixer paper. Examples are eq (1), eq (2). The plots are also hard to interpret, and more explanation could be better. The general structure of the paper could also be improved, to have a more coherent story. For example, section 3.2 could be merged with section 5.
2. The contribution is weak. For example, while it's good to formalize the relationship between MLP mixer and wide MLP, it's not that unexpected. From figure 1(d), it seems like the equivalent MLP under-performs the MLP-Mixer which makes the equivalence argument weak. The random permuted mixers also don't consistently out-perform the vanilla mixer which is also a weak argument.

**Questions:**

1. Do you have an explanation on why RP-Mixers are not consistently better than normal mixer.
2. the $vec(X)$ in eq(4) should be $vec(WX)$ and $vec(XV)$?

---

> ### Author Response · Authors · 2023-11-18
> **Response to Reviewer XJtr: Refinement of the structure of the paper and additional experiments**
>
> Thank you so much for your helpful suggestions.  Based on them, we have updated the manuscript.
> Our point-to-point responses to your comments are given below.
>
> > For one, a lot of notations are not defined, which requires the reader to find out from the original MLP mixer paper. Examples are eq (1), and eq (2).
> >
>
> Following your suggestion, we have added an explanation of MLP-Mixer in Section 2.2.   We have added some additional captions and refined the legend in Figure 1. Furthermore, we merged  Section 3.2 to Section 5 as long as possible.
>
> > For example, while it's good to formalize the relationship between MLP mixer and wide MLP, it's not that unexpected
> >
>
> As far as the authors know,  there is no literature that reveals any relationship between MLP-Mixer and standard MLP. Certainly, the transformation of the equation is simple, but we believe that being simple does not mean it lacks novelty.
>
> > From Figure 1(d), it seems like the equivalent MLP under-performs the MLP-Mixer which makes the equivalence argument weak.
> >
>
> It seems that our insufficient explanation might cause a misunderstanding of the purpose of Fig. 1(d).  The purpose is to understand  MLP-Mixer’s performance tendency based on width.  What we pointed out from Fig1(d) are the following two understandings:
>
> (1) MLP-Mixer has the same properties as a standard MLP at usual widths,
>
> (2) MLP-Mixer overcomes the disadvantage of MLPs that occur at extremely large widths. The disadvantage, reported by (Golveba et al., 2021), is from the eigenvalue distribution of weight matrices. This discussion is mathematically intricate, so it was delegated to the Appendix. Moreover, we add an explanation of the eigenvalue distribution in Section 4.2.
>
> In conclusion, we believe that an understanding of MLP-Mixer can be achieved by distinguishing between its similarities and slight differences from standard MLPs in Fig1(d).
>
> Furthermore, to emphasize the similarity in the usual width,  we have added the performance comparison for small width into Fig1(d).  The results make it more clearly show the similarity between MLP-Mixer and MLP in usual widths.
>
> > The random permuted mixers also don't consistently out-perform the vanilla mixer which is also a weak argument.
> &
> Do you have an explanation on why RP-Mixers are not consistently better than normal mixers?
> >
>
> Our insufficient explanation seems to have caused the reviewer’s confusion.  Our goal is to create an equivalent model with consistent performance, **essentially revealing that the performance of the Mixer is controlled by its width and sparseness.**  Therefore, RP-Mixer does *not* need to overcome the vanilla mixer.
>
> Let us also emphasize that RP-Mixer is not necessarily better than MLP-Mixer when the depth is limited. This is because random permutations break the structure for sharing tokens between mixing blocks. If we have a sufficiently large depth, we can overcome this decrease of performance as is confirmed in Figure 6.
>
> > the  vec(X) in eq(4) should be vec(WX) and vec(XV)?
> >
>
> The equation represents maps and vec(X) represents the source element of the map.  However,  as the Reviewer said it may be not easy to follow, so we have rewritten it into matrix form in the revised version. Thank you for your suggestion.
>
> If our replies have resolved the unclear points you raised, we would greatly appreciate a re-evaluation of the score.

---

> > ### Author Response · Authors · 2023-11-22
> > **Further comments  before author response period ends?**
> >
> > Thank you once more for dedicating your time to review our paper. As the discussion period nears its conclusion, we wish to ensure that our responses have adequately addressed your primary concerns. Please feel free to share any additional comments or queries; we are eager to respond to them.

---

### Author Response · Authors · 2023-11-22
**Rebuttal Sumary**

We are grateful for the valuable feedback from all reviewers. We are glad that **b7KH** found our work “nobel” and “experiments are well done.”, and  **Xjtr** for recognizing our work “this is the first time the equivalence between MLP mixer and wide MLP has been formalized.”

In response to the reviewers' suggestions, we conducted additional experiments, leading to an **enhanced understanding of MLP-Mixer**. Key updates during the rebuttal period include:

- The inclusion of a **computational resource analysis**, presented in Table 1 and Section 4.2. This analysis demonstrates that the RP-Mixer is significantly more efficient than the equivalent sparse and wide MLP in terms of memory efficiency, FLOPs, and runtime.
- The restructuring of Section 3.2 and Section 5 for improved readability, aiding comprehension of our experiments on network widening.
- Clarifications in the concept of MLP-Mixer to facilitate understanding without needing to refer back to the original MLP-Mixer paper.

---

### Meta-Review · Area_Chair_7sKu · 2023-12-14

**Metareview:**

Reviewers while find the papers analysis interesting, they find the contribution to be limited and the paper lacking clear takeaways. Authors response helped clarify the papers motivations, but overall it is unclear for reviewers what to takeway from the observation that you can rewrite Mixer as a wide MLP. Widening the analysis to other architectures, or showing more demonstrations and experiments can help the paper. I recommend rejection for the current draft.

**Justification For Why Not Higher Score:**

Lack of clear takeaways

**Justification For Why Not Lower Score:**

N/A

---

### Decision · Program_Chairs · 2024-01-16

Reject